# Grit (effortful persistence) can be measured with a short scale, shows little variation across socio-demographic subgroups, and is associated with career success and career engagement

**Clemens M. Lechner**[1]*, **Daniel Danner**[2], **Beatrice Rammstedt**[1]

**1** Department of Survey Design and Methodology, GESIS–Leibniz Institute for the Social Sciences, Mannheim, Germany, **2** University of Applied Labour Studies, Mannheim, Germany

\* clemens.lechner@gesis.org

**Data Availability Statement:** Researchers interested in replicating the results may apply to the Research Data Centre PIAAC to obtain the above-mentioned data sets: PIAAC; http://dx.doi.

## Abstract

Grit (effortful persistence) has received considerable attention as a personality trait relevant for success and performance. However, critics have questioned grit's construct validity and criterion validity. Here we report on two studies that contribute to the debate surrounding the grit construct. Study 1 ($N$ = 6,230) examined the psychometric properties of a five-item grit scale, covering mainly the *perseverance* facet, in a large and representative sample of German adults. Moreover, it investigated the distribution of grit across sociodemographic subgroups (age groups, genders, educational strata, employment statuses). Multiple-group measurement models demonstrated that grit showed full metric, but only partial scalar, invariance across all sociodemographic subgroups. Sociodemographic differences in the levels of grit emerged for age, education, and employment status but were generally small. Study 2 investigated how grit relates to career success (income, job prestige, job satisfaction) and career engagement (working overtime, participation in continuing professional development courses, attitudes toward lifelong learning) in an employed subsample ($n$ = 2,246). When modeled as a first-order factor, grit was incrementally associated with all indicators of career success and especially of career engagement (.08 ≤ β ≤ .75)—over and above cognitive ability and sociodemographic characteristics. When modeled as a residual facet of conscientiousness, grit largely retained its criterion validity for success but only partly for engagement (–.14 ≤ β ≤ .61). Our findings offer qualified support for the psychometric quality of the short grit scale and suggest that grit may provide some added value in predicting career outcomes. We critically discuss these findings while highlighting that grit hardly differs from established facets of conscientiousness such as industriousness/perseverance.

org/10.4232/1.12660; PIAAC-L; http://dx.doi.org/10.4232/1.12925.

**Funding:** This paper was supported by a grant from the German Federal Ministry of Education (BMBF; https://www.bmbf.de/en/index.html) to BR and DD [grant number 323 – 21381 – PEB]. The publication of this article was supported by the Open Access Fund of the Leibniz Association. The funders had no role in study design, data collection and analysis, decision to publish, or preparation of the manuscript.

**Competing interests:** The authors have declared that no competing interests exist.

## Introduction

The personality trait "grit" denotes the disposition to pursue long-term goals with sustained effort, zeal, and interest over time [1,2], or in short: *effortful persistence* [3,4]. Since its introduction a decade ago by Duckworth and colleagues [5], research on grit has attracted considerable, and often enthusiastic, media attention [6,7]. The compelling narrative that proponents of the grit construct have advanced [8] is readily summarized: People with higher grit work more strenuously to achieve their long-term goals, they persist in the face of setbacks or plateaus in progress, and they maintain their focus on these goals without being easily distracted by other (more short-term or less important) goals. For this reason, gritty individuals are more successful in achieving their goals and in attaining excellence in competitive environments. Obviously, this narrative resonates with popular beliefs about the value of hard work—beliefs that are succinctly captured in the aphorism "Winners never quit and quitters never win." Another reason why some researchers and practitioners have enthusiastically embraced grit is the hope that grit might offer a target for interventions aimed at fostering individual agency, performance, and success [9,10].

Despite the initial acclaim it received, the grit construct has also drawn considerable criticism in research [6] and popular media [7]. Perhaps most prominently, in their recent meta-analytic review of the grit literature, Credé, Tynan, and Harms [11] questioned grit's construct validity, casting doubt on its two-facet structure and its distinctness from conscientiousness. Moreover, they voiced concern that grit's criterion validity (especially that of its consistency of interest facet) vis-à-vis academic success had been exaggerated. They also pointed out that grit's criterion validity outside the academic context, on which most studies have focused, had yet to be demonstrated. Accordingly, these authors cautioned that interventions aimed at enhancing grit might be premature [6,11]. They called for more rigorous research into whether grit—especially its more promising perseverance facet—shapes success and performance.

Here, we briefly review the ongoing debate surrounding the grit construct and report on two empirical studies that contribute to this debate. Study 1 sheds light on the question as to how grit is distributed in the population and in sociodemographic subgroups. To that end, we present an in-depth investigation of the psychometric properties of a short five-item grit scale—capturing mainly the perseverance facet—in a large and representative sample of German adults. After testing measurement invariance across sociodemographic subgroups (age groups, genders, educational strata, and employment statuses), we examine potential differences in the levels of grit across these subgroups.

Study 2 then sheds light on the criterion validity of this short grit scale for career success (income, job prestige, job satisfaction) and engagement (working overtime, participation in continuing professional development courses, attitudes toward lifelong learning). Using a subsample of gainfully employed respondents from the same large-scale survey, we test whether grit is incrementally associated with these outcomes over and above cognitive ability and sociodemographic characteristics. We also illuminate the hotly debated [6,11] question as to whether grit possesses incremental validity over conscientiousness, of which grit is a facet [1,12]. Study 2 thus responds to recent calls for inquiries into the predictive power of grit for career outcomes [6,11,13].

## The debate about grit as a predictor of success and performance

### Differences in grit across demographic subgroups

One of the central questions in both the initial publication on grit [5] and the recent meta-analysis by Credé and colleagues [11] was whether levels of grit differ between

sociodemographic subgroups such as age groups, genders, educational attainment, or ethnicity. If grit is to be used for selection or placement purposes in educational and occupational contexts, small (or no) differences between subgroups would be desirable. Small differences would imply that grit is a resource that can be cultivated even among potentially disadvantaged groups, such as lower socioeconomic strata, and would reduce the risk of an adverse impact on legally protected groups such as minorities [11].

Previous research has indeed found only minor socio-demographic differences in the levels of grit. The only robust exception appears to be an increase in grit with age. A positive association of grit with age emerged already in initial publications [5], was replicated, for example, in an independent Japanese large-scale sample [14], and was also meta-analytically confirmed [11], even though it was weak ($\rho$ = .12). This relationship may reflect either age or cohort effects, although it has generally been interpreted as a sign of personality maturation [5]. Other initially reported differences, such as an educational gradient whereby higher-educated individuals reported higher grit, did not replicate in Credé and colleagues' meta-analysis [11].

The validity of such mean-level comparisons is predicated on the assumption that grit can be adequately assessed in all sociodemographic subgroups—in other words, that grit shows measurement invariance. For comparisons of grit levels across subgroups to be valid, scalar invariance (i.e., equal factor loadings and item intercepts) should hold [15]. Unfortunately, this psychometric prerequisite has not been routinely tested in the key studies on grit. For example, in their original publication introducing the construct, Duckworth and colleagues [5] compared the levels of grit across age segments and educational strata of their sample based on manifest scale scores without testing for measurement invariance [16]. The same is true for most of the studies on which Credé and colleagues' [11] meta-analysis was based. Although this does not necessarily invalidate these studies' findings concerning group differences, some caution is warranted when interpreting them because they might be biased by measurement non-invariance.

As a second prerequisite for testing differences between major sociodemographic subgroups, large and diverse samples in which the groups of interest are adequately represented are needed to detect—and properly quantify—differences between these groups. Few of the samples used in prior studies on group differences in grit meet this criterion. As a cursory glance at the appendix to Credé and colleagues' [11] meta-analysis reveals, the existing grit literature relies mostly on comparatively small ($N < 500$) and often highly selective samples (e.g., West Point Academy recruits; novice teachers; salespersons; spelling bee participants). Using such selective samples is consistent with the assumption that individual differences in grit are most consequential in highly challenging and competitive environments [5,8]. However, using selective samples runs the risks of non-representativeness, potentially low statistical power, and range restriction, rendering such samples less than ideal as a basis for testing sociodemographic differences in grit. Apart from this, the vast majority ($> 90\%$ in the meta-analysis) of the said samples are from the North American context, leaving open the question as to whether their findings concerning group differences in grit generalize to other world regions.

Thus, as it stands, the questions of (a) whether the psychometric properties of grit are equally good across major sociodemographic subgroups, and (b) how grit is distributed across these major sociodemographic subgroups in heterogeneous samples, has yet to be comprehensively answered. This applies particularly to non-U.S. samples. In Study 1, we will address these questions using a large and heterogeneous sample of adults from Germany.

## Does grit contribute to career success?

The widespread enthusiasm surrounding the grit construct is part of a broader trend in psychology [17,18] and economics [19] toward studying the power of personality or "character"

traits [4] in predicting important life outcomes. Indeed, most of grit's appeal as a construct appears to stem from its purported ability to predict the attainment of normative life goals—over and above cognitive ability, which researchers have traditionally credited with being a key, and often the strongest predictor of life success [20,21]. Some of the most widely-cited publications on grit [1,5,8] claimed that grit might be an equally potent, or even more potent, predictor of life success than cognitive ability, a claim that Duckworth has widely publicized through a TED talk [22]. These studies also presented evidence that grit was largely independent of cognitive ability. Whereas Credé and colleagues' [11] meta-analysis confirmed ($\rho$ = .05) the latter claim, current evidence does not support the former claim that grit is generally a *stronger* predictor of success and performance than cognitive ability is [6]. However, several studies found grit to be at least an *incremental* predictor of such outcomes over cognitive ability [4,5,16,23,24]. Such findings have nourished the hope that grit (and similar personality or character traits) might be an apt target for psychological interventions aimed at fostering individual agency [10,19], one that might be more malleable than cognitive ability especially at later ages [25].

Is this hope that cultivating grit might foster life success justified? Despite some research pointing to beneficial effects of grit for success outcomes, there are three caveats with regard to existing studies. First, in Credé and colleagues' [11] meta-analysis, grit did not fully live up to its promise of being a strong determinant of success, showing only modest associations with academic performance and retention. After adjusting for traditional measures of conscientiousness, the perseverance facet remained significantly associated with academic performance ($\beta$ = .25), but the consistency facet ($\beta$ = −.05) and overall grit ($\beta$ = −.02) did not. Conversely, traditional measures of conscientiousness did explain variance in these outcomes after first adjusting for grit. The authors recommended that future research on grit should focus on the perseverance facet.

Second, with some exceptions [1,3,14,23], the range of outcomes studied in extant grit research is largely confined to the academic realm. Akin to the original publications by Duckworth and colleagues [5,16], the majority of subsequent studies on the predictive power of grit have focused on academic success measures such as grade-point average (GPA) or retention in academic institutions. By contrast, few studies have investigated grit's criterion validity for success in other domains, including career success, as several scholars have remarked [6,11,13].

Third, the few studies that did address the criterion validity of grit for work and career outcomes yielded mixed results. For example, one study investigated how grit relates to retention and performance in novice teachers [26]. Observer ratings of grit constructed from the teachers' résumés prospectively predicted both their retention through the school year (*OR* = 2.34) and their (supervisor-rated) teaching effectiveness (*OR* = 1.60), whereas teachers' SAT scores and college GPA failed to predict these success criteria. Another recent study found that grit predicted sales representatives' retention over six months (*OR* = 1.38) over and above the Big Five and sales experience [1]. In a large sample of employed Japanese adults, grit was associated ($\beta$ = .47) with work engagement (assessed using a self-report scale) even after accounting for conscientiousness [14]. However, both grit and its two constituent facets were unrelated to income, a key indicator of career success, in the same sample. Another study outside the North American context investigated the validity of grit in predicting a range of workplace outcomes, including organizational citizenship behavior, in-role performance, counter-productive work behaviors, and job satisfaction, in a small sample of Romanian workers [13]. The authors found that grit showed only weak links to these outcomes, ranging from *r* = .04 for job satisfaction to *r* = .22 for in-role behavior. After accounting for the Big Five, the only statistically significant association of grit was to organizational citizenship behavior ($\beta$ = .50). Two further recent studies addressed the role of grit in (self-employed) entrepreneurs. The first study [27]

found entrepreneurs' grit levels to predict their venture's performance (as rated by an independent employee) one year later (β = .13). Along similar lines, study [28] in a sample of Austrian entrepreneurs found that perseverance was positively related to (self-rated) venture performance (β = .17), mainly through firm-level innovativeness. Consistency of interest was negatively related to innovativeness (β = −.14) but (marginally) positively related to success (β = .11). The only cross-nationally comparative study to date, using small and non-representative samples from 19 countries, found that grit was related to income (−.06 ≤ β ≤ .34) and job satisfaction (.02 ≤ β ≤ .35) only in a minority of the investigated countries after accounting for cognitive ability and education [23].

Overall, then, the link between grit and work or career outcomes remains tenuous. Although the literature yields some support for grit's utility in predicting success and performance, the evidence base is still small. Moreover, existing studies share important limitations, especially their reliance on mostly small, non-representative, and often highly selective sample (e.g., novice teachers, entrepreneurs, salespersons). In Study 2, we will investigate how grit relates to subjective and objective indicators of career success and engagement in a large and representative sample of German workers—above and beyond an extensive and high-quality measure of cognitive ability, education, and other socio-demographic characteristics.

### Does grit offer added value over conscientiousness?

Another recurring point of debate has been whether grit is sufficiently distinct from the Big Five dimension of conscientiousness. Duckworth and colleagues asserted grit's conceptual distinctness from global conscientiousness and its facets (especially self-control) by stressing grit's unique focus on *long-term* perseverance and consistency in goal pursuit, often over several years [1,5,29]. These authors found grit to predict success and performance after adjusting for conscientiousness [1,5], as did others [14].

Others have disputed that grit is distinct from Conscientiousness on conceptual and empirical grounds. Credé and colleagues' meta-analysis [11] concluded that grit was empirically indistinguishable from conscientiousness, with corrected correlation coefficients approaching unity. As noted above, their meta-analysis also found that grit hardly predicted academic success after adjusting for conscientiousness. Furthermore, studies using genetically sensitive designs suggest that grit and conscientiousness share largely the same similar genetic basis. In a large British twin study [30], the high genetic correlation between grit–perseverance and conscientiousness (r = .86) exceeded the phenotypical one (r = .53); grit also had little incremental criterion validity over conscientiousness. In another study, [4], grit loaded on a 69% heritable common "character" factor together with related constructs such as mastery and need for cognition. In view of such evidence, some commentators [6,7,11] have suggested that grit is indistinguishable from conscientiousness and hence a case of a "jangle fallacy" or "old wine in new bottles." According to this view, grit is simply a new label for conscientiousness that might help popularize the construct but has little to add to a longstanding research tradition.

An alternative view—to which Duckworth and colleagues have subscribed [1]—is that grit is a *facet* of conscientiousness, rather than being identical to conscientiousness as a whole [1,3,12]. According to this view, grit offers added value over general conscientiousness because grit emphasizes long-term goal striving—as opposed to facets such as orderliness, dutifulness, and self-control. Of course, even this more favorable view does not preclude that grit is a "jangle fallacy" because grit may not be sufficiently distinct from other conscientiousness facets that also emphasize longer-term goal striving or perseverance [11,12,31]. Both the definition of grit and a glimpse at the items from the grit scale (especially its perseverance facet) suggest

considerable overlap with the *achievement-striving* and *self-discipline* facets of conscientiousness in the Revised NEO Personality Inventory (NEO-PI-R ([32]), the *productiveness* facet of conscientiousness in the Big Five Inventory–2 (BFI–2 [33]), and the *industriousness* facet of conscientiousness described in in-depth studies on conscientiousness [31,34]. For example, one item measuring the productiveness facet of conscientiousness in the BFI–2 reads: "I am someone who is persistent, works until the task is finished" [33].

However, the criticism (which we believe is valid) that grit is almost indistinguishable from conscientiousness or at least some of its facets need not detract from the practical utility of the grit construct. Locating grit and attendant empirical findings in the Big Five framework enables researchers to fruitfully study grit by asking questions such as whether the grit facet is more predictive of success and performance than general conscientiousness or other conscientiousness facets. Several previous studies have judged the criterion validity of grit unfavorably on the grounds that grit failed to show incremental effects over and above conscientiousness in multiple regression [11,13,30]. However, if one conceives of grit as a facet of conscientiousness, controlling for conscientiousness (modeled as another first-order factor on the same level as grit or, more problematically [35], a manifest scale score) may not be the most desirable or conceptually sound approach. Rather, one should explicitly model the hierarchical structure of conscientiousness, whereby a grit facet is subordinate to the conscientiousness domain. Before doing so, one may wish to establish the predictive power of a grit facet as such, that is, irrespective of its being part of a higher-order conscientiousness domain (i.e., without controlling for conscientiousness). Treating grit as a facet that deserves to be studied in its own right would be in line with a recent trend toward studying facets (or even single items) as opposed to broad domains (for a discussion, see [17]). In Study 2, we pursued both these strategies, testing the criterion validity of grit modeled as a first-order factor in its own right and of grit modeled as a residual facet of conscientiousness.

## Study 1: Psychometric properties and distribution of a short grit scale

### Aims

Study 1 sought to shed light on the question as to how grit is distributed in the general adult population and across sociodemographic subgroups. Toward that end, we first conducted an in-depth evaluation of the psychometric properties of a short five-item grit scale in a large and diverse sample of German adults. Of particular interest was whether grit could be measured in the same way (i.e., showed measurement invariance) across subgroups defined by the most central and widely studied sociodemographic characteristics: age, gender, educational attainment, and employment status.

We then analyzed differences in the levels of grit across these sociodemographic subgroups. Our aim in so doing was to replicate and extend evidence on potential group differences in grit. Of particular interest were age differences in grit, the only clear-cut sociodemographic difference to emerge from previous research [11].

### Method

**Data.** Data for Study 1 came from the first wave (2014) of the PIAAC Longitudinal Study (PIAAC-L; [36]), a follow-up to the cross-sectional 2012 Programme for the International Assessment of Adult Competencies (PIAAC) in Germany. The PIAAC sample comprised adults aged 16 to 65 years who were randomly selected from local population registers in randomly selected German municipalities. At the end of the interview, PIAAC respondents were

asked whether they were willing to be re-contacted for a follow-up study (i.e., PIAAC-L) in the future. A total of 3,758 (or 69%) of the original 5,465 PIAAC 2012 respondents consented to be re-interviewed and could be successfully contacted for the follow-up in 2014. In addition, to these "anchor persons", their household members aged 18 and older were invited to participate, resulting in a total sample of 6,230 respondents from 3,758 different households. Trained interviewers from a professional survey institute conducted the computer-assisted personal interviews (CAPI) between March and August 2014. Participation was voluntary and incentivized by offering 25 euros to anchor persons and 10 euros to household members. The response rate was 72%. Approval through an ethics committee or review board was not required for PIAAC and PIAAC-L in Germany. The contracted survey institute is member of the European Society for Opinion and Marketing Research (ESOMAR) and complies fully with its standards (for details, see SOM). For more details on the sample and procedures, see the technical report to PIAAC-L [37].

**Measures.** The 2014 wave of PIAAC-L assessed grit with five of the eight items from the Short Grit Scale (Grit–S [16]). This scale originally posits two dimensions, perseverance of effort and consistency of interest. However, items from both facets are often combined into an overall grit score [1,5]. Moreover, as noted earlier, Credé and colleagues' meta-analysis [11] cast doubt on grit's two-dimensional structure and suggested that the perseverance facet predicted academic success, whereas the consistency of interest facet did not. The items selected for inclusion in PIAAC-L comprise all four items from the Grit-S scale measuring the more criterion-valid perseverance facet and one item measuring the consistency facet. The five items read as follows: (1) "I am a hard worker," (2) "I am diligent," (3) "I can cope with setbacks," (4) "I finish whatever I begin," and, from the consistency facet, (5) "I have difficulty maintaining my focus on projects that take more than a few months to complete." Experts in scale development and cross-national research from the PIAAC research team translated all the items into German (two separate translations followed by a reconciliation). In the original scale developed by Duckworth and colleagues, this item read "Setbacks don't discourage me". In order to avoid stringing together two negations, the German translation in PIAAC-L was positively worded ("I can cope with setbacks" / "Ich komme mit Rückschlägen gut zurecht"). Respondents were asked to rate the extent to which each statement applied to them on a fully labelled five-point scale ranging from 1 (*not at all*) to 5 (*to a very large extent*). We expected all items to form a single grit dimension that would reflect mainly the perseverance facet.

Our tests of measurement invariance and of differences in the levels of grit across sociodemographic subgroups required splitting the continuous age variables and recoding the education variable in a way that resulted in meaningful, large-enough, and roughly even-sized groups. We split age in years into three groups: young adults who are typically in their initial career stage (17 to 29 years; $n$ = 1481 or 23.8%), mid-aged adults in the prime working age (30 to 49 years; $n$ = 2457 or 39.4%), and older adults in their late career or retirement (50 years and older; $n$ = 2,293 or 36.8%). We coded gender such that men formed the reference group (1 = *female*, $n$ = 3,178 or 41%; 0 = *male*, $n$ = 3,053 or 49%). We coded educational attainment into three groups according to the level of the highest educational qualification obtained. We used the standard classification of educational attainment for the Comparative Analysis of Social Mobility in Industrial Nations (CASMIN) [38]. Educational attainment is a key dimension of social stratification that strongly determines life chances, especially in the context of the German educational system with its traditionally high importance placed on formal educational certificates [39]. We distinguished between lower, vocationally oriented education with 9–10 years of schooling (CASMIN levels 1–3; $n$ = 1,618 or 26.9%); intermediate, mostly vocationally oriented education and apprenticeships with typically 10–13 years of schooling (CASMIN levels 4–7; $n$ = 3,067 or 49.2%); and higher, academic/tertiary education (CASMIN levels 8–9;

*n* = 1,339 or 21.5%). We assigned a missing value to those who were still at school at the time of assessment (*n* = 163 or 2.62%). With regard to employment status, we distinguished between respondents who were currently employed at least 20 hours per week (i.e., full-time or part-time) and those who were not employed (1 = *employed*, *n* = 3,884 or 63%; 0 = *not employed*, *n* = 2,281 or 37%). S1 Table in the Supporting Information provides descriptive statistics for all variables used in Study 1. S2 Table shows their zero-order correlations.

**Analyses.** Our analyses comprised two main steps. In the first step, we tested a series of confirmatory factor analyses (CFA) in Mplus 8.0 to examine the psychometric properties of a unidimensional measurement model for the five-item grit scale. We began with a single-group model and proceeded to multiple-group models, testing the measurement invariance of grit across the major sociodemographic subgroups. We identified the model by fixing the variance of the latent grit factor to 1 and freely estimated all loadings, allowing for a full test of loading invariance. We used a robust maximum likelihood estimator (MLR) in conjunction with a sandwich-type estimator ("type = complex" in Mplus). This estimation method adjusts test statistics and standard errors for potential non-normality and corrects standard errors for the clustering (i.e., non-independence) of individual respondents in households. Although the rating scales were ordinal in nature, simulation studies show that treating items with five or more response categories as quasi-continuous yields essentially the same results as categorical estimators would [40]. We handled the small amount of missing data (see S1 Table) with the full information maximum likelihood (FIML) algorithm, which makes use of all available information and yields unbiased estimates under the assumption that data are missing at random (MAR). Even if data are not MAR, FIML typically results in less biased estimates than listwise deletion [41]. In line with current conventions for judging model fit [42,43], we chiefly relied on the comparative fit index (CFI), root mean square error of approximation (RMSEA) and the standardized root mean square residual (SRMR) to assess model fit. We judged model fit to be acceptable according to the following criteria: CFI and TLI > .90 ("adequate") or > .95 ("good"), RMSEA < .06, and SRMR < .09.

In the second step, we used the best-fitting measurement model to investigate the distribution of grit in sociodemographic subgroups. For this purpose, we estimated simple effects of all sociodemographic factors on grit to gauge the influence of age, gender, education, and employment on the mean levels of grit. Moreover, to investigate detailed age profiles of grit across sociodemographic subgroups, we fit locally weighted scatterplot smoothing (LOESS) curves on the factor score estimates derived from the best-fitting measurement model. This technique allows for a non-parametric and informative description of age differences in grit across sociodemographic segments.

## Results and discussion

**Measurement model.** Table 1 (Model 1) shows the fit of the measurement model for the full sample. The model showed good fit according to conventional criteria. An inspection of the normalized residuals (*z*) to detect local misspecifications suggested that the remaining misfit of the otherwise well-fitting model emerged from the covariances of the fifth grit item with the third (*z* = –3.74) and fourth (*z* = –4.47). Similarly, model modification indices suggested that the inclusion of these residual covariances might slightly improve model fit, although only slightly. For reasons of parsimony, we did not include any residual covariances.

An inspection of the standardized loadings revealed that the second item ("I am diligent"; λ = .72) had the highest loading on the grit factor, followed by the first ("I am a hard worker"; λ = .57) and fourth ("I finish whatever I begin"; λ = .56) item. Loadings of the third ("I can cope with setbacks"; λ = .36) and fifth ("I have difficulty maintaining focus. . ."; λ = −.34) item

**Table 1. Fit indices of the single-factor grit model.**

| Model | $\chi^2$ | df | CFI | TLI | aBIC | RMSEA | SRMR |
|---|---|---|---|---|---|---|---|
| 1. *Full sample* | 76.16 | 5 | .974 | .948 | 72593.65 | .048 | .022 |
| 2. *Grit by age group* | | | | | | | |
| a. configural | 107.95 | 15 | .967 | .934 | 72491.78 | .055 | .026 |
| b. metric | 129.67 | 23 | .962 | .950 | 72474.94 | .047 | .036 |
| c. scalar | 226.16 | 31 | .930 | .932 | 72535.16 | .055 | .046 |
| 3. *Grit by gender* | | | | | | | |
| a. configural | 79.04 | 10 | .974 | .949 | 72463.35 | .047 | .022 |
| b. metric | 81.56 | 14 | .975 | .964 | 72443.52 | .039 | .023 |
| c. scalar | 233.80 | 18 | .920 | .911 | 72595.14 | .062 | .046 |
| 4. *Grit by educational attainment* | | | | | | | |
| a. configural | 87.71 | 15 | .973 | .945 | 69899.72 | .049 | .023 |
| b. metric | 110.67 | 23 | .967 | .957 | 69883.90 | .044 | .034 |
| c. scalar | 191.10 | 31 | .940 | .942 | 69927.19 | .051 | .048 |
| 5. *Grit by employment status* | | | | | | | |
| a. configural | 93.73 | 10 | .968 | .935 | 71178.12 | .052 | .024 |
| b. metric | 98.67 | 14 | .967 | .953 | 71163.39 | .044 | .029 |
| c. scalar | 297.53 | 18 | .892 | .880 | 71366.44 | .071 | .046 |
| 6. *MIMIC models* | | | | | | | |
| a. no direct paths | 572.50 | 25 | .855 | .797 | 69076.28 | .061 | .032 |
| b. with direct paths | 139.30 | 20 | .968 | .945 | 68649.38 | .032 | .016 |

were much (and, as per pairwise comparisons through Wald tests, significantly) smaller than those of the other three and fell below commonly accepted thresholds for acceptable loadings (e.g., $\lambda \geq .40$). This means that the latent grit variable mainly reflected aspects that are similar to the established industriousness/productiveness facets of conscientiousness (i.e., being hard-working, diligent, finishing one's tasks; [31,33]). From the loadings and uniquenesses, we calculated average variance extracted (AVE; [44]). AVE expresses the average amount of variance in each item that is explained by the common factor. With $AVE = .27$, the amount of variance explained by the common grit factor was rather low on average.

We used coefficient omega $\omega$ [45] to estimate the reliability of the grit scale. Omega expresses the proportion of variance in the manifest scale score that the latent variable can account for. With $\omega = .63$, $CI_{90\%} = [.61, .64]$, reliability was below what most researchers would consider adequate, even for short scales. These reliability ($\omega$) and AVE estimates suggest that the manifest scale score should not be used. Latent-variable models that account for (un-)reliability should be used to predict outcomes of interest, lest bias arise in testing incremental criterion validity [35].

In sum, despite the overall good model fit, only three of the five items showed substantial loadings. The low AVE and omega estimates also suggest that the short grit scale is in need of further improvement. Moreover, they call for using latent-variable models that use only the reliable portion of variance, rather than manifest scale scores, for substantive analyses.

**Measurement invariance.** Models 2–5 in Table 1 show the results of our measurement invariance tests across sociodemographic subgroups (age group, gender, educational attainment, and employment status). For each sociodemographic variable, we compared a configural model that imposes no equality constraints on the loadings and intercepts across groups to a metric invariance model (i.e., factor loadings constrained to be equal across groups) and a scalar invariance model (i.e., loadings and intercepts constrained across groups). Metric

invariance ensures that the meaning of the latent construct is identical across groups. At least partial scalar invariance is required to test differences of latent means across groups [15]. We compared the fit of these models using the differences in goodness of fit (ΔGOF), $\Delta\chi^2$, and the sample-size adjusted Bayesian Information Criterion (aBIC). Regarding ΔGOF, we followed the simulation-based guidelines proposed by Chen [46], which stipulate that differences of ΔCFI ≥ .010, ΔRMSEA ≥ .015, ΔSRMR ≥ .030 when moving from a configural to a metric invariance model suggest loading non-invariance, whereas differences of ΔCFI ≥ .010, ΔRMSEA ≥ .015, ΔSRMR ≥ .010 suggest intercept non-invariance when comparing scalar to metric invariance. Regarding aBIC, lower values indicate a better balance between model fit and complexity (or parsimony).

According to these criteria, factor loadings were invariant across all sociodemographic groups considered. Specifically, when moving from the configural to the metric model, model fit did not deteriorate more than the cut-offs allow for the age groups model, $\Delta\chi^2(8) = 22.26$, $p = .004$, ΔCFI = −.005; ΔRMSEA = −.008, ΔSRMR = .010. The same was true for educational attainment, $\Delta\chi^2(8) = 22.96$, $p = .003$; ΔCFI = −.006; ΔRMSEA = −.005, ΔSRMR = .011. It also applied, with even a non-significant $\chi^2$ difference, to the models for gender $\Delta\chi^2(4) = 2.52$, $p = .64$, ΔCFI = .001, ΔRMSEA = −.008, ΔSRMR = .001; and employment status, $\Delta\chi^2(4) = 4.94$, $p = .29$; ΔCFI = −.001; ΔRMSEA = −.008, ΔSRMR = .005. Across the board, aBIC also favored the metric over the configural models.

By contrast, at least some item intercepts were non-invariant across groups. When moving from the metric to the scalar model, model fit worsened for age, $\Delta\chi^2(8) = 96.49$, $p < .001$, ΔCFI = −.032; ΔRMSEA = .008, ΔSRMR = .01; gender, $\Delta\chi^2(4) = 152.24$, $p < .001$, ΔCFI = −.055; ΔRMSEA = .023, ΔSRMR = .023; educational attainment, $\Delta\chi^2(8) = 80.43$, $p < .001$, ΔCFI = −.027; ΔRMSEA = .007, ΔSRMR = .014; and employment status, $\Delta\chi^2(4) = 196.86$, $p < .001$, ΔCFI = −.075; ΔRMSEA = .027, ΔSRMR = .017. The aBIC also favored the metric over the scalar models.

An inspection of the modification indices revealed that the lack of scalar invariance was due to only a few of the items: The intercept of the first item ("I am a hard worker") was non-invariant across age groups and employment status; that of the third item ("I can cope with set-backs") was non-invariant across the genders; and that of the fifth item ("I have difficulty maintaining focus . . .") was non-invariant across educational strata. All other constellations of items and sociodemographic factors showed no signs of intercept non-invariance.

In sum, these results show that metric invariance holds, ensuring that the meaning of the grit factor is the same across all four sociodemographic groups. However, scalar invariance did not hold. At least some item intercepts differed across groups. In order to avoid potential bias, this non-invariance should be taken into account when comparing the levels of grit across these sociodemographic groups.

**Socio-demographic gradients in the levels of grit.** Next, we turned to the question of how grit is distributed in the German adult population and across socio-demographic sub-groups. Our objectives were (1) to obtain unbiased estimates of mean-level differences by taking into account the non-invariance of the intercepts, and (2) to jointly model the effects of age, gender, education, and employment on grit in order to determine which of these sociodemographic factors is responsible for differences in grit after accounting for all others (this is the strategy pursued by Duckworth and colleagues in their initial publication on grit [5]).

Multiple-indicator-multiple-cause (MIMIC) models are often the method of choice for modeling intercept non-invariance. MIMIC models achieve this by allowing for direct effects on the non-invariant indicators. To assess group differences in the levels of grit, we thus estimated a MIMIC model in which we included the sociodemographic variables (coded with dummy variables) as predictors of the latent grit variable. In addition, we included direct

**Table 2. MIMIC results: Sociodemographic differences in grit and its items' intercepts.**

| Predictor | Direct effects | | | Grit (latent) |
|---|---|---|---|---|
| | Item 1[a] | Item3[b] | Item 5[c] | |
| Age group (ref.: 17 to 29 years) | | | | |
| 30 to 49 years | −0.00 | | | 0.03 |
| | (0.03) | | | (0.05) |
| 50 years and older | −.19*** | | | 0.10* |
| | (0.03) | | | (0.05) |
| Gender (ref.: male) | | −.30*** | | −0.02 |
| | | (0.02) | | (0.03) |
| Educational attainment (ref.: basic) | | | | |
| intermediate | | | −.14*** | 0.03 |
| | | | (0.03) | (0.04) |
| high | | | −.31*** | 0.16** |
| | | | (0.04) | (0.05) |
| Employed (ref.: non-employed | .34*** | | | 0.33*** |
| | | | | (0.04) |
| R$^2$ | | | | 0.03 |

[a]Item 1 = "I am a hard worker."

[b]Item 2 = "I can cope with setbacks."

\[c]Item 3 = "I have difficulties focusing. . ."

effects of these sociodemographic variables on some of the item intercepts that had proven non-invariant. We specified a direct effect of each sociodemographic variable on the item whose intercept received the highest modification index (i.e., the highest amount of intercept non-invariance) in the scalar invariance model with the respective sociodemographic variable as a grouping variable: of age group and employment status on the first item ("I am a hard worker"); of gender on the third item ("I can cope with setbacks"); and of education, coded with two dummy variables, on the fifth item ("I have difficulty maintaining focus. . .").

A MIMIC model with these four direct effects showed good fit to the data (Table 1, Model 6b)—and much better fit than a model including only the effects of the sociodemographic variables on the grit factor but no direct paths to the indicator (Table 1, Model 6a). As Table 2 shows, all direct effects were statistically highly significant, and some were substantial in size. The good fit of the model and the small model modification indices suggested that no additional direct effects were necessary.

Hence, we used Model 6b as a well-specified basis for probing group differences in grit. The paths from the sociodemographic variables to the latent grit factor in Model 6b, shown in the last column of Table 2, speak to sociodemographic gradients in grit after taking into account the non-invariance of the three item intercepts. These paths indicate the effects of each sociodemographic variable net of all others. There were three statistically significant differences: Grit was higher in the oldest age group, among more highly-educated respondents, and among the employed. Employment status produced the largest difference in grit, amounting to one-third of a standard deviation even after accounting for the intercept non-invariance in the "hard worker" item. There were no gender differences in grit (after accounting for males' higher intercepts on the "I can cope with setbacks" item).

In sum, our MIMIC model replicated the well-known increase in grit with age [11]. Additionally, it revealed higher levels of grit among those with higher education and especially

among the employed compared to the non-employed. However, sociodemographic gradients in grit were small, with all variables together explaining only 3.2% of the variance in grit.

**Age profiles of grit.** To further zoom in on the age profiles of grit, we computed the LOESS curves shown in Fig 1. These curves show how levels of grit differ by age in years in the full sample and when splitting the sample by the other sociodemographic variables. We constructed these curves based on the grit factor score estimates from Model 6b. Factor score determinacy was .83. S1 Fig in the Supporting Information provides the same graphs based on the manifest scale scores in the original metric of the items.

The age–grit relationship was curvilinear, suggesting that grit first slightly increases with age, reaches its peak by mid-adulthood, and then decreases again in older age (Fig 1, panel A). An examination of these age profiles separately by gender (panel B) suggested that, among older respondents (> 50 years), men score slightly higher than women. Respondents with tertiary education reported somewhat higher levels of grit, particularly in mid-adulthood, but less so at younger and older ages (panel C). The difference in grit between the employed and the non-employed was evident across the full age range (panel D). The apparent age-related increase in the levels of grit among the employed might reflect selection and causation. Those who stay employed at older ages may do so because of higher levels of grit in the first place; in turn, compared to those who are already retired, older employed individuals may have a higher need, as well as more frequent opportunities, to display gritty behavior.

In sum, these age profiles suggest that grit's (small) association with age is curvilinear, and that grit's age profiles differ somewhat across different sociodemographic segments. Future

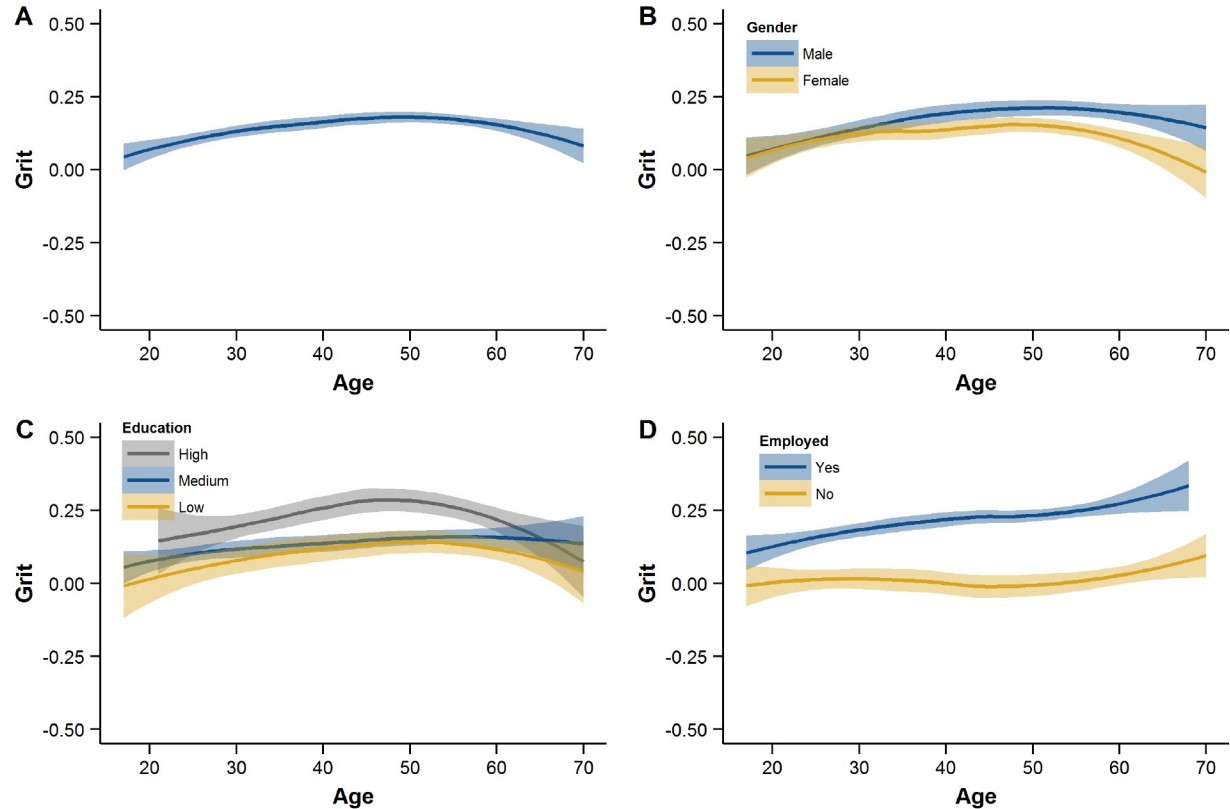

**Fig 1. Age profiles of grit.** LOESS curves (A) for the full sample, (B) by gender, (C) educational attainment, and (D) employment status. Values on the Y-axis are standardized factor scores.

research should unravel the extent to which these age profiles reflect age effects (e.g., age-grade-specific demands and opportunities to be gritty) or cohort effects (e.g., different socialization conditions instilling differential levels of grit), a question we could not resolve with our data.

## Study 2: Associations of grit with career success and engagement

### Aims

One of the central claims of grit's proponents was that grit explains individual differences in success and performance—above and beyond, and possibly better than, conscientiousness and even cognitive ability do [5,7,8]. The real touchstone by which to judge the grit's utility is, hence, its (incremental) criterion validity. However, few studies investigated grit's criterion validity for career outcomes (e.g., [1,3,14,23]). Accordingly, the question we asked in Study 2 is whether the short grit scale in PIAAC-L, covering mainly the perseverance facet, is incrementally related to career success and career engagement over and above cognitive ability, educational attainment, and other sociodemographic characteristics—and whether grit's effect sizes would indeed rival or surpass those of cognitive ability.

We also addressed the issue of whether the predictive power of grit persists after accounting for its being a facet of conscientiousness. To this end, we used recent extensions of bifactor models [47] to residualize grit for conscientiousness (see Method). If grit is conceived of as a facet of conscientiousness, as our literature review shows it should be, this modeling approach is conceptually more appropriate than controlling for conscientiousness in the traditional multiple-regression sense. By using latent variable models accounting for the (un-)reliability of grit, conscientiousness, and cognitive ability, we circumvented the risk of spurious incremental validity claims that plague studies that rely on manifest scale scores [35].

To test grit's criterion validity in relation to career outcomes, we selected three widely used [48] indicators of global career success: income and job prestige as objective indicators and job satisfaction as a subjective indicator. We selected working overtime and participation in continuing professional development (CPD) courses as objective indicators and respondents' attitude toward lifelong learning (hereafter referred to as "learning orientation") as a subjective indicator of career engagement. These indicators reflect the investment of time and effort that respondents are willing to make proactively in order to advance their careers. Based on previous studies, we hypothesized that grit would be incrementally associated with all six indicators of success and engagement. We expected that grit would generally be more strongly associated with engagement than with success because engagement depends more directly on the person and her traits than success does.

### Method

**Data.**   For Study 2, we used a subsample comprising all panelists from the 2014 wave of PIAAC-L [36] who had already participated in the 2012 PIAAC study (referred to as "anchor persons" in PIAAC-L) and were gainfully employed (at least 20 hours per week) at the time of the interview in 2014. For these 2,246 respondents, we matched data on cognitive ability from PIAAC 2012 with data on grit, conscientiousness, and career success and motivation from PIAAC-L 2014. Compared to the original PIAAC sample, the PIAAC-L sample was slightly biased towards higher education and younger age [37,49]. To correct any sampling biases, the PIAAC-L data distribution provides cross-sectional post-stratification weights for the anchor persons. These weights adjust the marginal distribution of key sociodemographic characteristics (age, gender, education, region, household size, and population of municipality) to that of the adult population in Germany (as taken from the German Microcensus 2012), resulting in a

sample that is representative of workers in German with regard to these sociodemographic characteristics. Because, in contrast to Study 1, there was only one respondent per household (i.e., the anchor person), there was no need to adjust standard error for clustering as in Study 1.

S3 Table in the Supporting Information shows descriptive statistics for all variables used in Study 2, described next. S4 Table provides their zero-order correlations.

**Measures.** Income, job prestige, and job satisfaction served as our indicators of career success. To measure income, respondents were asked to report their gross monthly earnings (i.e., regular employment income without special payments such as additional vacation pay) in the previous month in euros; we took the natural logarithm of their responses to obtain a more normally distributed income variable. To measure job prestige, we used Treiman's Standard International Occupational Prestige Scale (SIOPS) scores [50], which were generated from respondents' current occupations. SIOPS is among the most widely used measures of occupational prestige. It reflects the popular evaluation of occupational standing and is based on respondent ratings from numerous countries. SIOPS scores have a theoretical range from 0–100; average ratings for professions range from 12 (e.g., cleaner) to 78 (medical doctor). To measure job satisfaction, respondents were requested to indicate how satisfied they were with their job on an 11-point scale (0 = *completely dissatisfied*; 10 = *completely satisfied*).

Working overtime, CPD participation, and learning orientation served as our measures of career engagement. To measure working overtime, respondents were asked to indicate the number of hours they had worked overtime in the past month. To measure participation in CPD courses, respondents were asked to report how many CPD courses they had taken in the preceding year (i.e., in 2013). To measure their attitude toward lifelong learning, respondents were asked to indicate on a 7-point scale (1 = *do not agree at all*; 7 = *fully agree*) whether they personally agreed with the following statement: "In today's world of work, it is imperative to update, refresh, and broaden knowledge through further training and education".

Our focal independent variables in Study 2 were grit and—with the intent of comparing its effect sizes to those of perhaps the most well-established predictors of career success—conscientiousness, and cognitive ability. To measure *grit*, we used the five-item scale (four from the perseverance facet, one from the consistency facet) investigated in Study 1. We modeled grit as a unidimensional latent variable as in Study 1.

*Conscientiousness* was measured in the 2014 wave of PIAAC-L with a short three-item scale from the BFI-S [51]. Respondents were asked to indicate on a 7-point scale (1 = *does not apply at all*; 7 = *fully applies*) the extent to which each of the following statements applied to them personally: "I am someone who works thoroughly"; "I am someone who is rather lazy"; and "I am someone who carries out duties efficiently." As to be expected for a scale this short, reliability was rather low ($\omega$ = .56). However, supplemental analyses in an independent sample of German adults (*n* = 353) available to us (for more information, see S1 Appendix) showed that the short BFI–S conscientiousness scale correlates very highly (manifest *r* = .73; latent *r* = 1.0) with a full 12-item conscientiousness scale from the BFI–2 [33]. Interestingly, BFI–S global conscientiousness correlates more strongly with the productiveness/industriousness facet of the BFI–2 (manifest *r* = .72; latent *r* = .99) than with the responsibility/reliability (manifest *r* = .58; latent *r* = .90) and organization/orderliness (manifest *r* = .56; latent *r* = .75) facets. Together, these results suggest that the short BFI–S conscientiousness measure covers general conscientiousness adequately while being most closely related to the productiveness/industriousness facet. Because this facet is closest to grit in definition, controlling for the BFI–S conscientiousness measure in our analyses amounts to a conservative test of grit's incremental criterion validity over conscientiousness.

*Cognitive ability* was measured in PIAAC 2012 with three competence tests: literacy, numeracy, and problem solving in technology-rich environments (for detailed information on the tests, see [52,53]; for information the tests' implementation in the present dataset, see [37,54]). Literacy (58 items in total) refers to the ability to understand, use, and interpret written texts. Numeracy (56 items) refers to the ability to use, apply, interpret, and communicate mathematical information and ideas. Problem solving in technology-rich environment (14 items) captures the ability to successfully use digital technologies, communication tools and networks to search for, communicate and interpret information. All three tests were assessed using a multistage adaptive testing design. Test items were devised and extensively validated by an international commission of eminent scholars. Tasks were designed to reflect daily-life situations which respondents were typically highly motivated to solve. Moreover, interviewers were thoroughly trained for the assessment, they were present while respondents took the tests, and monitored the process. Although this was not a high-stakes test situation and the tests reflect typical rather than maximal performance, these steps ensured that respondents took the test situation seriously, and this is indeed what debriefings from the interviews suggested (personal communication from the German national project management team). For each skill domain, the PIAAC/PIAAC-L data distribution includes 10 sets of plausible values (PV) per respondent. We ran each of our models involving cognitive ability (described below) separately on each of the 10 sets of plausible values and aggregated the results while correcting the standard errors [55,56].

In the terminology of the updated Cattell–Horn–Carroll (CHC) theory of intelligence, the three PIAAC tests measure broad skill domains on Stratum II [57]. As such, their common variance provides a good indicator of general intelligence (*G*), which we obtained by modeling a latent *G*-factor across the three tests. As to be expected from the three subtests' high correlations in this sample ($.77 \leq r \leq .87$; see S4 Table), the reliability of cognitive ability was very high ($\omega = .94$). In subsequent measurement models, loadings on the common *G*-factor were accordingly high and homogeneous ($.88 \leq \lambda \leq .96$).

We controlled for sociodemographic characteristics that may be associated with both grit (see Study 1) and with career success and engagement, which renders them potential confounders: age in years (i.e., as a continuous variable); gender (1 = *female*, 0 = *male*); educational attainment, coded in the same way as in Study 1 (i.e., two dummy variables for intermediate and higher education, with lower education serving as the reference group); and type of employment contract (1 = *full-time*; 0 = *part-time*).

**Analyses.** We analyzed the relationships between grit and the four indicators of career success and motivation in two different ways, illustrated in Fig 2. In Model A, we modeled grit as a first-order latent variable and tested whether it predicted career success and motivation over and above cognitive ability and the sociodemographic controls (age, gender, educational attainment, type of employment). In Model B, we modeled grit as a facet of conscientiousness to test its criterion validity over and above conscientiousness (as well as over and above cognitive ability and the sociodemographic controls). Specifically, Model B is a Bifactor-(S–1) model [47] in which the three conscientiousness items and the five grit items all load on a conscientiousness factor; whereas the five grit items, but not the three conscientiousness items, load on a grit facet factor. This renders the conscientiousness factor a *reference* factor reflecting general conscientiousness as measured with the three BFI–S conscientiousness items (note that three items are sufficient to capture a reference factor; Eid and colleagues [58] also used three items to measure the reference factor in their illustrative models). The grit facet factor is then a *residual* factor that is orthogonal to the conscientiousness factor. It captures the specific variance in the grit items that these items do not share with the conscientiousness items. We identified the conscientiousness factor by fixing its loading on the third item ("I am someone

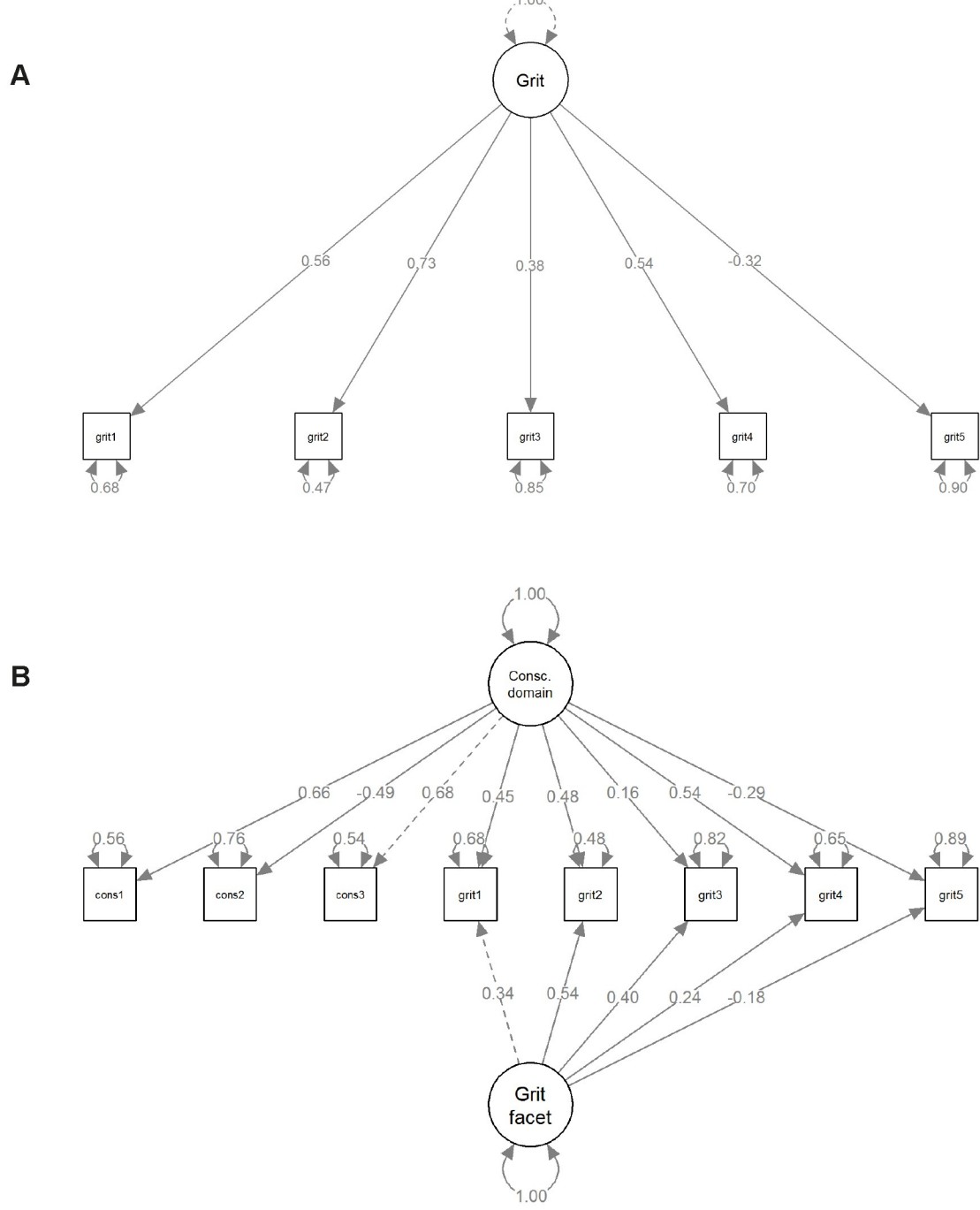

**Fig 2. Measurement models.** Alternative measurement models for grit as a first-order factor (Model A) and for grit as a residual facet of conscientiousness (Model B). All values are standardized parameters.

who carries out duties efficiently") to 1, and we identified the grit (facet) factor by fixing its loading on the first item ("I am a hard worker") to 1 and its mean to zero.

When comparing the subsequent regression results obtained from these two alternative models, it is important to bear in mind that the Model A and Model B answer distinct research

questions, and that the substantive meaning of grit is different in each. In Model A, grit is a first-order factor, and its interpretation as a measurement-error-free latent variable is straightforward. This model speaks to the criterion validity vis-à-vis career success and engagement of grit, taken "as such" and deliberately disregarding its overlap with conscientiousness. In Model B, grit is a residual facet of conscientiousness containing only the portion of variance the five grit items do not share with the three conscientiousness items. This model speaks to the question of whether the grit facet is incrementally associated with career success and engagement over and above conscientiousness.

If grit is best conceived of as a facet of conscientiousness [1,7], Model B is conceptually more appropriate than the typically used alternative model in which both grit and conscientiousness would be modeled as first-order factors or manifest scale scores in multiple regression. For the sake of completeness, we estimated a model in which grit and conscientiousness were correlated first-order factors. Such a model had borderline fit, $\chi^2(19) = 155.282$, $p = .000$, CFI = .926, TLI .89, SRMR = .039, aBIC = 44257.228. The correlation between grit and conscientiousness in this model was very high, $r = 0.78$, questioning the distinctness of both constructs. We did not consider further this conceptually inadequate model for our analyses.

## Results and discussion

**Measurement models.**  Fig 2 shows the standardized model parameters and fit indices for the first-order grit factor (panel A) and the Bifactor-(S−1) model (panel B). Both models also included the latent cognitive ability factor (omitted for simplicity).

Model A showed good fit to the data, $\chi^2(19) = 116.97$, $p < .001$, RMSEA = .048, CFI = .979, TLI = .968, SRMR = .042 ($M$ across 10 PV for each fit index). The pattern of loadings was virtually indistinguishable from the one obtained in the full sample in Study 1, as was the reliability of the grit scale score ($\omega = .63$).

Model B also showed a good fit to the data, $\chi^2(37) = 215.51$, $p < .001$, RMSEA = .046, CFI = .968, TLI = .953, SRMR = .035 ($M$ across 10 PV for each fit index). As shown in Fig 2, both the three conscientiousness items and the three grit items that are conceptually closest to the industriousness/productiveness facet of conscientiousness had substantial loadings on the conscientiousness reference factor. Loadings on the grit facet factor were mostly small. Only the second item ("I am diligent") and the third item ("I can cope with setbacks") had substantial loadings. Thus, the meaning of the grit facet factor in Model B is chiefly defined by these two items, aligning it more closely with the definitions of grit as effortful persistence over longer periods of time and in the face of setbacks. A higher score on this factor means that a person has higher grit (especially higher self-discipline and a better ability to cope with setbacks) than would be expected on the basis of his or her overall conscientiousness. The variance of the grit facet factor, Var(*Grit*) = 0.08, amounted to one-fifth of the conscientiousness domain factor, Var(*Consc.*) = 0.40. The grit facet factor explained an additional 28% of the variance in the five-item grit scale score beyond the 36% explained by the conscientiousness domain factor. These values were obtained by calculating the reliability coefficients omega ($\omega$) and omega hierarchical ($\omega_h$) for nested-factor models [59].

In the Bifactor-(S−1) model [58], the specificity coefficient reflects the share of variance in the true score of an item $\tau_{ik}$ (here: each of the five grit items) that can be uniquely attributed to a specific factor $\zeta_{ik}$ (here: the grit facet factor) and is not shared with the reference domain (here: the first conscientiousness item). It is calculated as

$$Spe(\tau_{ik}) = \frac{\lambda_{ik}^2 Var(\zeta_{ik})}{Var(\tau_{ik})}$$

whereby $\lambda_{ik}$ refer to the loadings on the specific ("facet") factor and $Var(\zeta_{ik})$ refers to the variance of that factor. In terms of specificity, the third grit item ("I can cope with setbacks") showed the highest specificity, $Spe(\tau_3) = .85$, indicating that 85% of its reliable (true-score) variance was grit-facet variance not shared with conscientiousness, followed by the second item ("I am diligent"), $Spe(\tau_2) = .55$. The fourth item ("I finish whatever I begin") had the lowest specificity, $Spe(\tau_4) = .16$, indicating that only 16% of its true score variance can be attributed to the grit facet, whereas 84% of its reliable (true score) variance could be attributed to the conscientiousness domain factor. For the first item ("I am a hard worker") and the fifth item ("I have difficulty maintaining focus. . ."), specificities were .36 and .28, respectively.

Finally, we examined associations of grit with cognitive ability. The first-order grit factor (Model A) was completely unrelated to cognitive ability, $r = -.003$, $p = .40$, 95% CI [−.11, .04]. This is in line with the preponderance of evidence [4,11] and buttresses the idea that grit is a resource that can be cultivated independently of (highly heritable) ability or talent [5]. By contrast, the grit residual facet factor (Model B) was positively related to cognitive ability, $r = .19$, $p = .004$, 95% CI [.06, .31]. The conscientiousness domain factor was negatively related to cognitive ability, $r = -.19$, $p = .004$, 95% CI = [−.26,−.12]. This negative relationship between conscientiousness and cognitive ability frequently emerges in large-scale studies, including in German adolescents and adults [60,61]. Contrariwise, the positive relationship of the grit facet factor diverges from grit's meta-analytically confirmed orthogonality to cognitive ability [11] but aligns with the positive relationships recently reported from a large-scale twin study that included a broad range of high-quality cognitive ability measures [4].

In sum, both measurement models showed a good fit in the employed subsample. Model B demonstrated that three of the five grit items loaded strongly on the conscientiousness domain, leaving only a small portion of variance to be captured by the grit facet factor. Nonetheless, the grit facet factor captured a unique portion of variance in the grit items that these items do not share with conscientiousness, and this portion was largest for the items referring to persistence while pursuing goals (being diligent and coping with setbacks). Results from both models confirmed that grit was independent of cognitive ability (Model A), apart from a small positive association of the residual grit facet factor (Model B).

**Criterion validity of the first-order grit factor.** Turning to our focal research question: How much can grit add to explaining career success and engagement? To answer this question, we regressed the six indicators of career success and engagement on the latent grit variable as in Model A (i.e., with grit as a first-order factor). Fig 3 (panel A) shows the fully standardized regression coefficients (β) with 95% confidence intervals of grit and, for comparison, of cognitive ability. These coefficients express by how many standard deviations (SD) each career outcome would change if grit and cognitive ability changed by 1 *SD*. Table 3 provides detailed regression results including all covariates. Coefficients in Table 3 are unstandardized for the sociodemographic covariates and *X*-standardized for grit and cognitive ability. Thus, coefficients in Table 3 can be interpreted as the effects of a one-unit change in each sociodemographic variable and a 1 *SD* change in grit and cognitive ability on each career outcome in its original metric. Recall that income was logarithmically transformed, and that coefficients for working overtime and participation in CPD courses are negative binomial regression coefficients.

As expected, the first-order grit factor was positively associated with all six indicators of career success and career engagement over and above all covariates in the model. But how large and relevant are grit's effect sizes? For better interpretability, we exponentiated the regression coefficients for income, overtime, and CPD course participation such that the coefficients for income can be interpreted as the income ratio of a person scoring +1 *SD* in grit to a person scoring at the mean of grit (i.e., zero), and the coefficients for overtime and CPD course

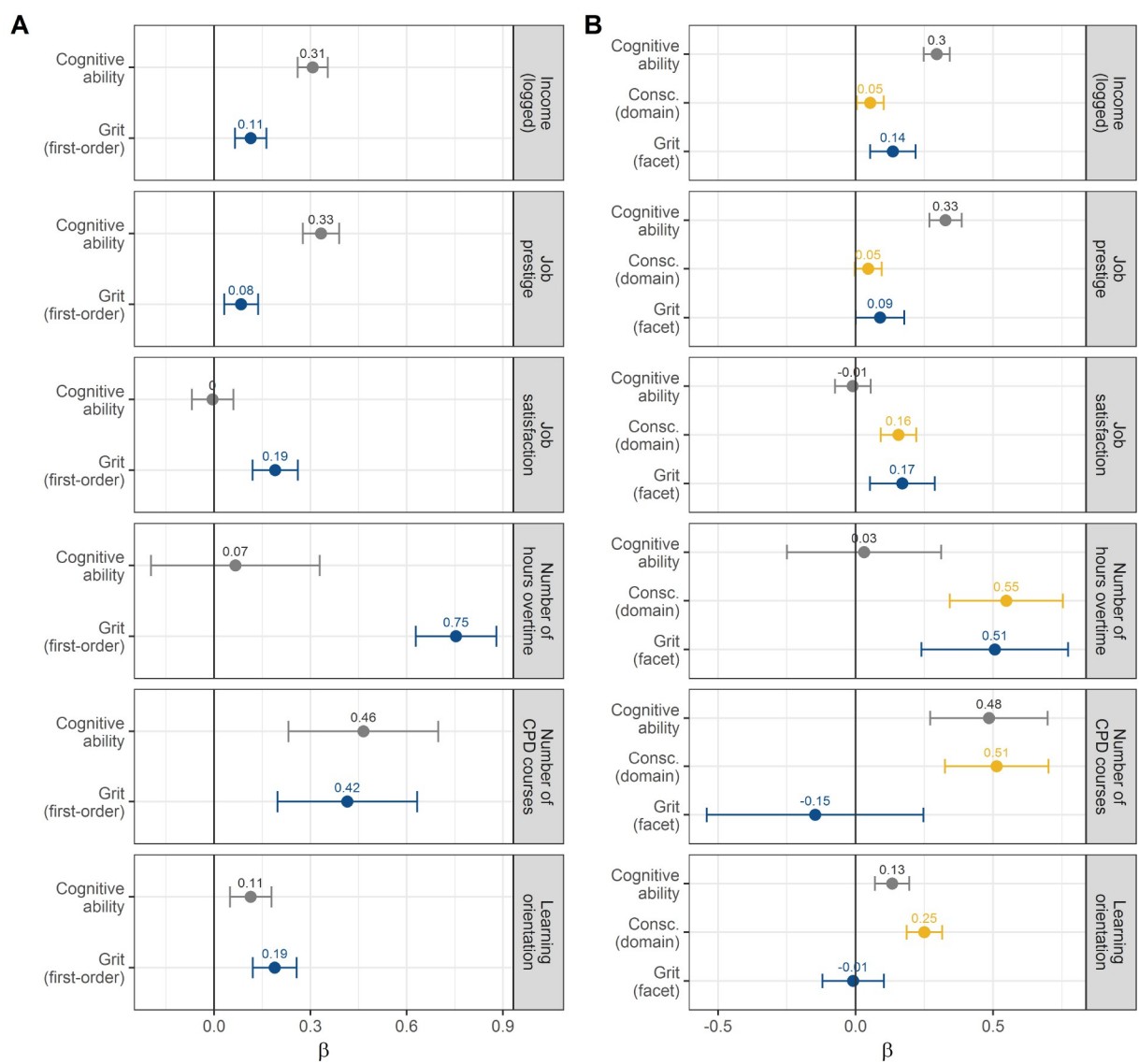

**Fig 3. Associations of grit with career success and career engagement.** (A) Associations for grit modeled as a first-order factor. (B) Associations for grit modeled as a facet of conscientiousness. For comparison, the associations of cognitive ability and conscientiousness with the same outcomes are shown. Points represent standardized regression coefficients (β). The lines represent 95% confidence intervals. All associations are controlled for the covariates shown in Tables 3 and 4.

participation can be interpreted as incident rate ratios (IRR). Compared to a person with an average level of grit, a person scoring +1 *SD* in grit would be expected to earn an 8.3% higher monthly income ($e^{0.08} = 1.083$). At the median monthly gross income of 2,500 euros (about 3000 USD at the time of the study) in this sample, this would be equivalent to 208 euros. Furthermore, he or she would be expected to score 1.15 points higher on the job prestige (SIOPS) scale, which equals 0.09 *SD* of this scale; and to score 0.38 points higher on the 11-point job satisfaction item, which equals 0.19 *SD*. In terms of career engagement, a person scoring +1 *SD* in grit would be expected to work 31% more hours overtime per week; to participate in 20% more CPD courses; and to score 0.16 points higher on the 7-point item measuring attitude toward lifelong learning, which equals 0.19 standard deviations in this measure. This

**Table 3. Grit (first-order factor) and career success and engagement: detailed regression results.**

| | Career Success | | | Career Engagement | | |
|---|---|---|---|---|---|---|
| | Income (logged) | Job prestige | Job satisfaction | Hours overtime | CPD participation | Learning attitude |
| Grit | 0.08*** | 1.15** | 0.38*** | 0.27*** | 0.18** | 0.16*** |
| | [0.04, 0.12] | [0.42, 1.88] | [0.25, 0.52] | [0.19, 0.34] | [0.07, 0.29] | [0.10, 0.22] |
| Cognitive ability | 0.22*** | 4.58*** | −0.01 | 0.02 | 0.21** | 0.10** |
| | [0.18, 0.25] | [3.76, 5.40] | [−0.14, 0.12] | [−0.07, 0.12] | [0.09, 0.32] | [0.04, 0.15] |
| Age in years | 0.01*** | 0.05* | 0.00 | −0.01 | 0.01** | 0.00 |
| | [0.01, 0.02] | [0.01, 0.10] | [−0.01, 0.01] | [-0.01, 0.00] | [0.00, 0.02] | [0.00, 0.01] |
| Female[a] | −0.11*** | 2.21*** | 0.05 | −0.22** | 0.25* | 0.07 |
| | [−0.17, −0.05] | [1.00, 3.42] | [−0.16, 0.26] | [−0.36, −0.08] | [0.06, 0.44] | [−0.01, 0.16] |
| Education[b] | | | | | | |
| Intermediate | −0.06 | 1.97* | −0.04 | −0.05 | 0.29* | 0.09 |
| | [−0.12, 0.01] | [0.36, 3.57] | [−0.31, 0.23] | [−0.23, 0.13] | [0.03, 0.54] | [−0.03, 0.20] |
| Higher | 0.19*** | 12.53*** | −0.13 | 0.29** | 0.56*** | 0.12 |
| | [0.10, 0.28] | [10.49, 14.56] | [−0.46, 0.20] | [0.07, 0.51] | [0.28, 0.84] | [−0.01, 0.25] |
| Full-time[c] | 0.73*** | 0.21 | −0.25 | 0.13 | 0.09 | −0.01 |
| | [0.66, 0.80] | [−1.19, 1.60] | [−0.50, 0.00] | [−0.04, 0.30] | [−0.12, 0.30] | [−0.10, 0.09] |
| $R^2$ | .48 | .38 | .04 | .08[d] | .05[d] | .06 |

Coefficients for the two latent variables (grit and cognitive ability) are standardized with regard to *X*; all other coefficients are unstandardized. Standard errors in parentheses.

[a]reference = male.

[b]reference = lower (CASMIN level 1–3).

[c]reference = part-time employed.

[d]Pseudo-$R^2$ calculated from a linear model.

***$p < 0.001$

**$p < 0.01$

*$p < 0.05$.

pattern suggests that grit is more strongly associated with career engagement than with career success, as we expected.

How did the predictive power of grit compare to that of cognitive ability, which is widely credited with being the strongest predictor of life success [20,21,60]. The first-order grit factor in Model A had about three times weaker associations with income and prestige than cognitive ability had. It had roughly equally sized associations as cognitive ability with the number of CPD courses taken and the association to lifelong learning, but stronger associations than cognitive ability with job satisfaction and especially with working overtime.

As per the $R^2$ values in Table 3, the model explained a substantial share of the variance in objective career success—almost half the variance in income and more than a third in job prestige. By contrast, the models explained less than ten percent in the other four outcomes.

In sum, these regression results support the incremental criterion validity of grit, modeled as a first-order factor, in relation to career success and career engagement—over and above cognitive ability, educational attainment, and other sociodemographic factors. Effect sizes were small to moderate. At the same time, our findings contradict the claim that grit is a more potent predictor of success than cognitive ability, at least, as far as objective dimensions of success (income, prestige) are concerned and at least in a broad population sample (as opposed to selective samples from highly challenging and competitive environments).

**Criterion validity of the grit facet factor.** How, then, does grit relate to career success and engagement when applying a Bifactor-(S–1) model in which grit is a residual facet of conscientiousness? Fig 3 (panel B) shows the standardized regression coefficients (β) with 95% confidence intervals for grit compared to conscientiousness and cognitive ability. Table 4 provides *X*-standardized for grit, conscientiousness, and cognitive ability and unstandardized regression results for the other covariates.

The grit facet factor was incrementally associated with income, job prestige (although the 95% CI bordered zero), job satisfaction, and especially working overtime, over and above all other covariates in the model. Compared to a person with an average level of grit, a person scoring +1 *SD* in the grit facet would be expected to earn a 10% higher monthly income, the equivalent of about 249 euros; have a job with a 1.2 points higher prestige score (0.09 *SD*); to score 0.34 points higher on the job satisfaction scale (0.17 *SD*); and to work 21% more hours overtime. The grit facet factor was unrelated participation in CPD and learning orientation. Thus, the grit facet factor in Model B was somewhat less strongly, and less consistently, associated with the career outcomes than the first-order grit factor in Model A.

How does the criterion validity of the grit facet factor compare to those of conscientiousness and cognitive ability? The conscientiousness reference factor was positively associated with working overtime, participation in CPD courses, and attitudes toward lifelong learning; as well as with income and job satisfaction. Overall, conscientiousness was less consistently related to career success than the residual grit facet—but more consistently related to career engagement. Again, cognitive ability outpredicted both grit and conscientiousness with regard to income and prestige; it was also associated with CPD course participation and learning orientation but not with working overtime.

In sum, these results show that grit, when modeled as a residual facet of conscientiousness, largely retained its criterion validity for career success. The grit facet hardly predicted engagement, whereas conscientiousness did. Even though grit's effect sizes were small and grit did not generally outpredict cognitive ability and conscientiousness, this suggests that the five grit items capture something more than the three conscientiousness items—something that may be relevant to career success.

## General discussion

What are the ingredients of success in education, at work, and beyond? This question continues to intrigue laypersons and personality psychologists alike. With its promise to be one such ingredient, and perhaps a vital one that even outranks cognitive ability [1,8,26], the construct of grit (i.e., effortful persistence over long periods of time) has recently gained traction. However, grit has also polarized the field. Grit's critics have questioned the construct validity and its distinctness from conscientiousness and have pointed to its limited (incremental) criterion validity [6,7,11,13,30].

Our two studies contribute several novel insights to the debate on the utility of the grit construct. Study 1 offered an in-depth psychometric validation of a short grit scale in a large and diverse sample of German adults from the PIAAC-L survey. This grit scale comprises all four perseverance item plus one consistency item from the Grit–S scale [16]. We know of only one prior study that validated a grit scale in Germany [24]. However, that study used smaller and more selective samples and tested measurement invariance only for gender. Study 1's findings show that grit can be adequately measured with a unidimensional model in all sociodemographic segments we examined (age groups, gender, educational attainment, and employment status). Overall, the psychometric quality of the scale was acceptable, with some important qualifications. First, two of the items had only small loadings; consequently, the meaning of

**Table 4. Grit (residual facet factor) and career success and engagement: detailed regression results.**

| | Career Success | | | Career Engagement | | |
|---|---|---|---|---|---|---|
| | Income (logged) | Job prestige | Job satisfaction | Hours overtime | CPD participation | Learning attitude |
| Grit (facet factor) | 0.10** | 1.22* | 0.34** | 0.21*** | −0.07 | −0.04 |
| | [0.04, 0.15] | [0.01, 2.44] | [0.10, 0.59] | [0.10, 0.32] | [−0.30, 0.17] | [-0.15, 0.07] |
| Conscientiousness (reference factor) | 0.04* | 0.63 | 0.32*** | 0.17*** | 0.25*** | 0.22*** |
| | [0.00, 0.07] | [−0.05, 1.31] | [0.19, 0.45] | [0.10, 0.24] | [0.12, 0.37] | [0.16, 0.28] |
| Cognitive ability | 0.21*** | 4.49*** | −0.02 | 0.02 | 0.23*** | 0.11*** |
| | [0.17, 0.24] | [3.65, 5.32] | [−0.15, 0.11] | [−0.08, 0.11] | [0.10, 0.35] | [0.06, 0.17] |
| Age in years | 0.01*** | 0.05* | −0.00 | −0.01 | 0.01* | 0.00 |
| | [0.01, 0.02] | [0.01, 0.10] | [−0.01, 0.01] | [−0.01, 0.00] | [0.00, 0.02] | [0.00, 0.01] |
| Female[a] | −0.07* | 2.62*** | 0.12 | −0.21** | 0.15 | −0.02 |
| | [−0.13, −0.01] | [1.20, 4.04] | [−0.14, 0.38] | [−0.37, −0.06] | [−0.07, 0.36] | [−0.11, 0.08] |
| Education[b] | | | | | | |
| Intermediate | −0.08* | 1.75* | −0.08 | −0.05 | 0.35** | 0.13* |
| | [−0.14, −0.01] | [0.07, 3.43] | [−0.36, 0.20] | [−0.24, −0.13] | [0.09, 0.61] | [0.01, 0.25] |
| Higher | 0.15** | 12.14*** | −0.20 | 0.27* | 0.73*** | 0.23** |
| | [0.06, 0.25] | [9.89, 14.40] | [−0.57, 0.17] | [0.03, 0.50] | [0.40, 1.05] | [0.09, 0.38] |
| Full-time[c] | 0.73*** | 0.21 | −0.26* | 0.13 | 0.11 | −0.01 |
| | [0.66, 0.80] | [−1.20, 1.62] | [−0.51, −0.01] | [−0.04, 0.30] | [−0.10, 0.32] | [−0.11, 0.10] |
| $R^2$ | .49 | .38 | .05 | .08[d] | .05[d] | .08 |

Coefficients for the three latent variables (grit, Conscientiousness, and cognitive ability) are standardized with regard to $X$; all other coefficients are unstandardized.

Standard errors in parentheses.

[a] reference = male.

[b] reference = lower (CASMIN level 1–3).

[c] reference = part-time employed.

[d] Pseudo-$R^2$ calculated from a linear model.

***$p < 0.001$

**$p < 0.01$

*$p < 0.05$.

the latent grit variable was largely defined by items similar to industriousness/productiveness facet of conscientiousness (being hard-working, diligent, and finishing one's tasks)—and less so by being able to cope with setbacks and maintaining focus on long-term projects. Second, the reliability (internal consistency) of the scale was rather low. Moreover, although factor loadings were invariant, some item intercepts were non-invariant across some sociodemographic groups. This implies that the five-item grit scale can still be used for predictive purposes and/or to compare groups on grit, but appropriate latent-variable methods accounting for measurement error (i.e., unreliability) and intercept non-invariances should be used for such purposes.

Our analysis of how grit is distributed across socio-demographic subgroups corroborated previous conclusions that socio-demographic differences in the levels of grit are small [11]. Besides replicating a slight increase in grit with age [5,11] and educational attainment [5], we found higher levels of grit among the employed compared to the non-employed. Our more nuanced analyses of age differences further revealed that the age profile of grit is curvilinear rather than linear. Although our cross-sectional data cannot disentangle age from cohort effects, a curvilinear age trend in grit would be consistent with well-established models in

developmental psychology that suggest that people's capacity to actively pursue their goals peaks in mid-adulthood and then declines as biological aging takes its toll [62].

As we and others have argued [5,11], the crucible for judging the utility of the grit construct is ultimately its criterion validity. In Study 2, we, therefore, investigated whether grit is related to career success and engagement—above and beyond, and potentially more strongly than, conscientiousness and cognitive ability. Results supported for the criterion validity of grit vis-à-vis career outcomes. Grit, when modeled as a first-order factor, was positively related to all six objective and subjective indicators of career success and engagement—incrementally over cognitive ability and sociodemographic characteristics (age, gender, education, employment type). Whereas the effect sizes of grit were small for career success ($.08 \leq \beta \leq .19$), they were substantial for career engagement ($.19 \leq \beta \leq .75$)—sometimes exceeding those of cognitive ability, to which the first-order grit factor was unrelated as in most earlier research. It should be born in mind that a brief five-item grit scale competed here with a *G*-factor across three extensive cognitive tests; against this backdrop, grit's criterion validity can be judged favorably.

Somewhat ironically, the pattern of results was suggestive of a less-than-ideal balance between grit's effects on career engagement versus those on actual success: The extra hours that grittier people invest in working overtime and participating in CPD courses appear not to fully translate into a higher income or job prestige, at least when viewed from a cross-sectional perspective. Inasmuch as working overtime can incur costs (i.e., in terms of health risks and work–family conflict), grit's associations with career engagement may not be unequivocally beneficial. It is plausible to assume that grit influences career success through (sustained) engagement with career goals [63]; thus, it is possible that gritty individuals do reap the benefits of their heightened career engagement in the long run. Longitudinal studies are needed to trace whether grit's effects on career success are mediated through career engagement and how these effects unfold from a lifespan perspective. Nonetheless, the key message of Study 2 is that grittier people are somewhat more successful and certainly more engaged in their jobs.

But does grit, which has been scorned to be a "jangle fallacy" or "old wine in new bottles," offer any added value over traditional measures of conscientiousness in predicting career success and engagement? According to Study 2, grit partly retained its criterion validity when modeled as a residual facet of conscientiousness in a Bifactor-(S−1) model. Arguably, this constituted a conservative test of grit's incremental criterion validity over conscientiousness. This is because the three-item BFI–S conscientiousness measure emphasizes the industriousness/productiveness facet of conscientiousness to which grit is conceptually closest, leaving little unique variance for grit. Still, the grit facet factor was still positively associated with income, working overtime, and (less clearly) job satisfaction and prestige. Grit's effect sizes rivaled or exceeded those of conscientiousness for career success and working overtime but were smaller than those of conscientiousness for CPD course participation and learning attitude. Thus, neither the grit facet nor the conscientiousness domain was universally superior to the other in terms of criterion validity. As an anonymous reviewer noted, the fact that grit hardly predicted engagement after removing the variance it shares with conscientiousness may call into question whether what distinguishes the constructs is really grit's focus on long-term goal striving (i.e., engagement). The grit facet factor did not consistently outpredict cognitive ability, especially not for the objective career success measures (income and job prestige).

Overall, Study 2 lends qualified support to the criterion validity of grit vis-à-vis career outcomes: It largely confirmed that grit incrementally predicts career success and engagement and that it does so incrementally over cognitive ability and, at least partly, over conscientiousness. By contrast, Study 2 yields little support for the claim that grit is generally more important for success and performance than cognitive ability and traditional measures of conscientiousness [5,26].

Put into a broader perspective, our findings lend further support to the relevance of conscientiousness and its facets (of which grit is one), which are increasingly recognized as potent predictors of academic success and life success more broadly [18,34,64]. In view of the mostly correlational nature of extant evidence, it remains to be seen whether grit and its relatives from the conscientiousness family have a truly causal effect on academic and career success, and whether they are indeed amenable to interventions (for critical discussions, see [11,30,65]). Should interventions designed to enhance grit prove to be successful (see [9], for initial positive evidence from a randomized control trial), this would nourish the hope that individuals could be helped to unfold their potential by staying on track with their goals—that is, by being gritty.

### Limitations and directions for future research

Our findings have two main limitations. The first concerns the measures at our disposal in PIAAC–L. Because of time and questionnaire space constraints, both grit and conscientiousness were measured with short scales. This is a typical trade-off in multi-thematic large-scale surveys. Short scales are increasingly common and are often able to retain a considerable amount of the criterion validity of longer scales [66]. However, compared to longer scales, short scales are typically less reliable and sometimes content deficient, which can lead to attenuated and more variable criterion correlations [66–68]—although their criterion validity can sometimes exceed that of longer scales if the longer scale contains criterion-irrelevant items [66]. Our use of latent-variable models ensured that the grit scale's low reliability did not bias our conclusion. However, we were unable to recover the proposed two-facet structure of grit [5] in our analyses because PIAAC-L administered only five grit items—all four perseverance items from the Grit–S scale [16] but only one of its consistency items. Although the perseverance facet has emerged as more powerful predictor of success outcomes than the consistency facet or overall grit [11], future research using full-length grit scales that allow modeling and comparing both grit facets would be an important addition to our findings.

With the short three-item conscientiousness scale available in PIAAC-L, we were also unable to address in full detail the relationship of grit to conscientiousness. The three-item conscientiousness measure correlated very highly with a longer conscientiousness scale in an independent sample of German adults in our supplemental analyses. Thus, the three items offered a solid basis to estimate the conscientiousness reference factor to control for the variance grit shares with conscientiousness. The three items did not, however, allow us to model other established facets of conscientiousness as additional residual factors. Future studies using a more comprehensive measure of conscientiousness that covers several facets (e.g., the BFI–2 or NEO-PI-R) could conduct facet-level analyses comparing the grit to other conscientiousness facets such as productiveness/industriousness or orderliness/organization.

Apart from cognitive ability, all measures used in this study were self-report measures. As such, they are prone to response styles such as acquiescence and socially desirable responding that can introduce common method bias and distort model fit, criterion correlations, and other covariance-based statistics (e.g., [69]). Future research should analytically account for such response styles, such as by using balanced scales to counter acquiescence, and by observer-rated or objective measures of grit (e.g., [26]), conscientiousness, and career outcomes.

Second, because our data were correlational and cross-sectional, Study 2 was unable to ascertain the causal status of grit, a limitation it shares with the vast majority of studies on the predictive power of personality traits for life outcomes. Although we followed the common assumption in this literature that personality traits causally influence career success and

engagement, it is not entirely implausible that the reverse causal order also holds. For example, people might infer their self-reported levels of grit from their work engagement, such as working overtime and participating in CPD courses. Ultimately, only intervention studies in which grit is manipulated will yield conclusive evidence concerning its causal status [65]. Moreover, genetically informative design could be used to unravel the extent to which grit is shaped by environmental or genetic influences; and to what extent genetic or environmental variation in grit accounts for its associations with career success (for a recent application to academic achievement, see [4]).

## Conclusion

Critics have—in our view convincingly—argued that grit is a case of a "jangle fallacy" or "old wine in new bottles" [6,7,11]. We concur that grit is hardly different from established conscientiousness facets, especially industriousness/productiveness. Nonetheless, findings from our two studies suggest that the much-decried grit construct may hold some value for research on career success. Findings from Study 1 showed that grit can be measured equivalently (i.e., shows metric and partial scalar invariance) across major demographic segments of the population. Findings also confirmed that grit shows only small differences across socio-demographic subgroups (i.e., by age, education, and employment status). That said, we found the five-item grit scale in PIAAC-L to be in need of improvement and caution against using its manifest scale score. Findings from Study 2 supported grit's criterion validity. They showed that grit— even if measured with only five items—was incrementally associated with career success and especially with career engagement over and above cognitive ability. Grit largely retained its criterion validity for career success (but less so for engagement) after accounting for conscientiousness, of which grit is a facet. Additionally, results from Study 2 supported the idea that grit is a resource that is largely independent of cognitive ability

In view of these findings, we believe that grit can contribute to our understanding of career success and engagement. However, to fruitfully study grit, we strongly recommend that researchers take the Big Five as a frame of reference and conceive of grit as a facet of conscientiousness. Studying grit as a facet of conscientiousness would be consistent with a larger trend in the field toward studying narrower facets or even single items so as to maximize predictive power and obtain a clearer understanding of the mechanisms linking traits to outcomes [17]. Granted, if grit is simply a facet of conscientiousness, one could question whether future studies should continue to use measures of grit at all—or should instead employ well-validated measures of conscientiousness and its established facets, such as the BFI–2 [33], which contains a "productiveness" facet that is nearly indistinguishable from grit in definition and item wording. Jury on this subject is still out, and it is beyond the scope of our present contribution to resolve this question.

## Supporting information

**S1 Table. Descriptive Statistics for Study 1.**
(DOCX)

**S2 Table. Zero-Order Correlations for All Study Variables (Study 1).**
(DOCX)

**S3 Table. Descriptive Statistics for Study 2.**
(DOCX)

**S4 Table. Zero-Order Correlations for All Study Variables (Study 2).**
(DOCX)

**S1 Fig. Age profiles based on manifest mean scores.**
(DOCX)

**S1 Appendix. Validity of the Conscientiousness Instrument Used in Study 2.**
(DOCX)

## Author Contributions

**Conceptualization:** Clemens M. Lechner, Daniel Danner.

**Data curation:** Clemens M. Lechner.

**Formal analysis:** Clemens M. Lechner.

**Funding acquisition:** Daniel Danner, Beatrice Rammstedt.

**Investigation:** Clemens M. Lechner.

**Methodology:** Clemens M. Lechner, Daniel Danner.

**Project administration:** Clemens M. Lechner.

**Resources:** Clemens M. Lechner, Beatrice Rammstedt.

**Supervision:** Beatrice Rammstedt.

**Visualization:** Clemens M. Lechner.

**Writing – original draft:** Clemens M. Lechner.

**Writing – review & editing:** Clemens M. Lechner, Daniel Danner, Beatrice Rammstedt.

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
