## [Decision Letter · Decision Letter 0]

16 Aug 2019

PONE-D-19-18723

Can Grit Add to Our Understanding of Career Success? Psychometric Properties and Predictive Power for Career Success and Engagement of a Short Grit Scale

PLOS ONE

Dear Dr. Lechner,

Thank you for submitting your manuscript to PLOS ONE. After careful consideration, we feel that it has merit but does not fully meet PLOS ONE’s publication criteria as it currently stands. Therefore, we invite you to submit a revised version of the manuscript that addresses the points raised during the review process.

I was able to collect three reviews from three anonimous Reviewers. Please, see their comments appended at the bottom of this letter. As you will see in these specific comments, there were several major concerns with the current version of your study. After my own reading of the manusript, I think that this might be considered as a major review. Therefore, please notice that a resubmission will require an additional round of reviews. Besides, the final outcome of the process cannot be predicted at this point. If you decide to resubmit a revised version of your manuscript, please provide either a proper answer or rebuttal to each of the suggestions that were raised by the Reviewers.

We would appreciate receiving your revised manuscript by Sep 30 2019 11:59PM. To enhance the reproducibility of your results, we recommend that if applicable you deposit your laboratory protocols in protocols.io, where a protocol can be assigned its own identifier (DOI) such that it can be cited independently in the future. For instructions see: http://journals.plos.org/plosone/s/submission-guidelines#loc-laboratory-protocols

We look forward to receiving your revised manuscript.

Kind regards,

Angel Blanch, Ph.D.

Academic Editor

PLOS ONE

Journal Requirements:

Reviewers' comments:

Reviewer's Responses to Questions

**Comments to the Author**

1. Is the manuscript technically sound, and do the data support the conclusions?

Reviewer #1: Partly

Reviewer #2: Yes

Reviewer #3: Yes

2. Has the statistical analysis been performed appropriately and rigorously? 

Reviewer #1: Yes

Reviewer #2: Yes

Reviewer #3: Yes

3. Have the authors made all data underlying the findings in their manuscript fully available?

Reviewer #1: Yes

Reviewer #2: No

Reviewer #3: Yes

4. Is the manuscript presented in an intelligible fashion and written in standard English?

Reviewer #1: Yes

Reviewer #2: Yes

Reviewer #3: No

5. Review Comments to the Author

Reviewer #1: The manuscript proposed to be published is very interesting, both for what it intends to contribute and for the research and applications that can be derived from it. However, for this purpose, work should improve in various aspects. The most important ones are listed below:

The manuscript is too long. An article of almost 18,000 words is exceptional today, a fact that makes the article very hard to follow. Please, stick to the usual extensions of scientific contributions nowadays (between 5,000 and 6,000 words).

As regards the background, important but insufficient information is included. On the one hand, by not contemplating variables that, later in the analysis of results, are valued as very important. On the other hand, due to the lack of elaboration and integration between the different parts / paragraphs of the introduction. In this sense, for example, study 1 (L. 76) or the importance of sociodemographic differences (L. 90 onwards) and the objective of “added value of grit as a predictor of success and performance” are not sufficiently justified. (L. 74-75)

Moreover, in the background, other similar and opposite grit constructs should be considered, with the aim of highlighting points of convergence and divergence (for example, the concepts of perception of competence, vocation or procrastination).

In line with the above, the objective is very generic and the analysis of results addresses issues that are not related to this objective.

The results and discussion are considered together. The guidelines of the magazine propose that they constitute two distinct sections. Additionally, the length of these sections (7 pages for the first study and 9 for the second) makes reading and understanding the content very difficult.

Some statements are poorly justified, for example, L. 147, "key publications"... It would be convenient to explain the reason why they are considered "key".

The introduction and conclusions should further emphasize the applications of the results as well as the consequences thereof.

The writing of the work should be improved. Here are some problems that need to be corrected:

On several occasions, acronyms are used without making explicit what they refer to (PIAAC; CASMIN ...).

Line 283, a section with the title "Study 1" opens and the next line begins "Our aims in Study 2"

The first person of the plural is used, recommending the impersonal style.

L. 63, citations 6 and 7 present considerable criticism, but this criticism is of a very different nature (one is a scientific article and the other comes from mass media). Likewise, these citations that appear on line 50 should also be assessed for their relevance in the context where they appear. In short, it is suggested to quote with more precision and criticism.

The title of the work has been edited with words in capital letters.

The expression "Conscientiousness" is written throughout the text with the first capital letter. No other expression appears following this rule (?).

In summary, it is suggested to carefully rewrite the whole work, which may lead to a good article. Once the extension of the same is reduced, a review can be made more adjusted to the concreteness of what has been presented.

Reviewer #2: The following technical concerns should be considered in the revision:

- Problem of short scales: Given large-scale omnibus studies as PIAAC, there is now a strong effort to use short scales in the hope of transferring properties from the parent scale to the short scale. Due to the fact that the manuscript is methodological oriented (“measurement invariance”), the general problems of short scales should be discussed as well (e.g., Smith et al., 2000). Especially, I wonder why a short scale of GRIT with 5 items was applied instead of the original scale in this validation study at the expense of a possible low reliability of the scale.

- Structural equation modeling: The application of SEM was justified by low reliability (Omega=.63) of the scale (“Row 413 “…call for latent-variable models”). First, SEM is the method of choice in social sciences, if scales are prone to random measurement errors irrespective of the size of measurement errors. Second, reliability is a necessary, but not sufficient condition for validity despite any attenuation formulas. If the reliability is low, the validity is restricted in size. In my view this issues should be consider at least as part of the final discussion (limitations).

- Response sets, social desirability: Rating scales are prone to response sets (e.g., acquiescence, middle-scale) and effects of social desirability, for example, the item “I am hard worker”. Such biases were mentioned in the manuscript (row 578), but they are, unfortunately, not considered in the statistical analysis. The revised manuscript should give some reasons, why the statistical analysis did not account for these possible distortions driven by the question format (Boeckenholt, 2017).

- Measurement model: Eventually, concepts of classical test theory were used to establish the measurement model. Rating scales were used, which are at best ordinal scaled. I wonder whether measurement models considering the ordinal character of the data are more adequate than models based on continuous scales, for example, an ordinal Rasch model with a threshold concept. In my view the reasons to favor concept of classical test theory toward the probabilistic test theory is worth to be shortly discussed in the revised manuscript.

- Statistical model comparison: The model comparison was mainly based on the goodness of fit indices, as CFI and residual indices as RMSEA. I wonder why the so-called information criteria as BIC (mentioned as aBIC? in the manuscript) were not used for model comparisons. With BIC the scalar measurement model can be clearly ruled out (Table 1). Given the low inter-correlations among the variables (Table S4), it could be expected that most of the models fit the data rather well despite low reliability. Model fit and reliability do not have to match (“First, it is possible that the scale is simply too short.“ Stanley et al., 2016, p. 984).

-Minor: I would like to point out that the title is only understandable for readers, who know what GRIT is. Perhaps the underlying construct is also interesting for other readers.

References

Smith, G. T., McCarthy, D., M., Anderson, K. G., (2000). On the sins of short-form development. Psychological Assessment, 12(1), 102-111.

Boeckenholt, U. (2017). Measuring response styles in likert items. Psychological Methods, 22(1), 69-83.

Stanley, L. M., Edwards, M. C. (2016). Reliability and Model Fit, 76(6), 976–985

Reviewer #3: Thank you for the opportunity to review the interesting manuscript. The authors conducted extensive and rigorous analyses on large-scale data of the German population to test the reliability and validity of the grit construct. I admire the thoughtful and thorough analytic approach taken by the authors, as they contribute compelling evidence regarding the value of grit in relation to other related constructs. Below I offer some suggestions to potentially improve the manuscript from its current state.

1. For the categorical variables (gender, education, employment), please indicate the frequencies of participants under each category.

2. In their evaluations of the factor loadings of grit items (for example, lines 391-395 and 741-749), the authors report that some items had “smaller loadings” while others were “substantial” based on the readings of coefficients. To add objectivity to the inference, authors may want to consider statistically testing the (in)equality of the coefficients. I believe there are several approaches to do this, like the one discussed in this paper:

Kwan, J.L.Y. & Chan, W. (2011). Comparing standardized coefficients in structural equation modeling: a model reparameterization approach. Behavior Research Methods, 43(3), 730-745. https://doi.org/10.3758/s13428-011-0088-6

3. In addition to the two models displayed in Figure 2, how would the model fit if you include grit and Conscientiousness in the model as independent first-order factors and assume an inter-factor correlation. I would expect that the model’s fit will be lower than that of model 2B, but the comparison would contribute to demonstrating whether, and to what extent, grit should be considered as a construct independent from Conscientiousness or as a facet of Conscientiousness.

4. What specifically was the method of analyses for the results displayed in Figure 3? Were they simple regressions, or were they multiple regressions where you controlled for cognitive ability and conscientiousness when you regressed the DVs on grit?

5. In line 864, the authors report that the “grit factor was unrelated to job prestige,” while the result displayed in Table 4 indicate the particular coefficient was significant at least at the .05 level. Perhaps the authors were referring to the results of Figure 3, but it is difficult to evaluate the effects just from the visualization of the CIs. Please indicate specific CI values for the effects depicted in Figure 3.

6. According to Study 2, grit, when treated as a residual factor after controlling for Conscientiousness, retains its associations with variables such as income, job satisfaction, and hours spent working overtime, but ceases to predict two career engagement variables (i.e., number of CPD courses taken, learning orientation). I find this an interesting set of findings and wish for the authors to elaborate on what the results imply. For instance, the findings seem to contradict the existing view that grit differs from general Conscientiousness in that it “emphasizes long-term goal striving” (line 246 of manuscript). The results of the present study suggest that general Conscientiousness more strongly predicts employees’ orientation toward long-term career development, while grit holds predictive power over people’s engagement with the job at hand (i.e., how long they persist with the task, how affectively committed they are with the current job). I would like to see a discussion on what the findings add to the debate about how grit should be conceptualized.

Minor points:

7. Format for reference citation was inappropriate throughout the results section of Study 1, and in several other places as well (lines: 64, 375).

8. Unidentified footnotes appear in lines 48 and 149. Also please cite the TED talk mentioned in line 149.

9. Please check the sentence structure for lines 692-695.

10. Although the manuscript in overall was well-written, I found typographical errors throughout. Please thoroughly review the manuscript for errors. Below, I list ones I noticed with their line numbers:

95: such *as* lower socioeconomic strata / 183: grit was be associated / 285: Study 2 / 524: was .83. graphs but… / 560: how relates / 566: (3) / 599: these any / 626: though 827: pattern is suggests / 967: old wine in new *models / 992: wind out / 1004: helped use

6. PLOS authors have the option to publish the peer review history of their article (what does this mean?). If published, this will include your full peer review and any attached files.

Reviewer #1: Yes: Codina, N.

Reviewer #2: No

Reviewer #3: No

---

## [Author Response · Author response to Decision Letter 0]

30 Sep 2019

Reponses to Reviewer #1

[1.1] The manuscript proposed to be published is very interesting, both for what it intends to contribute and for the research and applications that can be derived from it. However, for this purpose, work should improve in various aspects. The most important ones are listed below:

Thank you for your positive overall evaluation of our manuscript. We are happy that you found it interesting. Your comments helped us in improving it further. Below, we detail how we responded to each comment.

[1.2] The manuscript is too long. An article of almost 18,000 words is exceptional today, a fact that makes the article very hard to follow. Please, stick to the usual extensions of scientific contributions nowadays (between 5,000 and 6,000 words).

We appreciate your concern about manuscript length. We did our best to shorten the manuscript while retaining all essential information—and while additionally incorporating the suggestions made by you and the other anonymous reviewers. Moreover, we restructured and rewrote some parts—especially the Results/Discussion sections—in order to improve their readability and clarity. The manuscript is now about 1,200 words (almost four pages) shorter. 

At the same time, we would like to emphasize that ours is a comprehensive two-study paper to which the word limit you are quoting (which we find somewhat arbitrary and which is not generally accepted) cannot be applied. Also, kindly note that PLOS ONE has no word limit. Its submission guidelines (https://journals.plos.org/plosone/s/submission-guidelines#loc-length) state that, “Manuscripts can be any length. There are no restrictions on word count, number of figures, or amount of supporting information.” 

Therefore, based on your comment, we aimed to present all findings more concisely and clearly, but we did not work toward any specific word limit. If, after this revision, you still deem the manuscript too long, we would be happy to shorten passages that you and other reviewers identify as overly lengthy, as well as to move material that you and other reviewers deem non-essential to the supplementary online material. 

[1.3] As regards the background, important but insufficient information is included. On the one hand, by not contemplating variables that, later in the analysis of results, are valued as very important. On the other hand, due to the lack of elaboration and integration between the different parts / paragraphs of the introduction. In this sense, for example, study 1 (L. 76) or the importance of sociodemographic differences (L. 90 onwards) and the objective of “added value of grit as a predictor of success and performance” are not sufficiently justified. (L. 74-75)

Following this suggestion, we now already mention the socio-demographic subgroups that we investigate in Study 1 in the Introduction (L. 76 onwards). We also list the specific outcomes of Study 2 (L. 80 onwards). Furthermore, we now explain the structure of our two-study paper better by stating more clearly how the two studies fit together and fit to the literature review. The improved introductory section reads as follows:

“Here, we review the ongoing debate about grit as a predictor of success and performance and report on two empirical studies that contribute to this debate. Study 1 sheds light on the questions as to how grit is distributed in the population and in sociodemographic subgroups. To that end, we conduct an in-depth investigation of the psychometric properties of a short five-item grit scale—capturing mainly the perseverance facet—in a representative sample of German adults. After testing measurement invariance across key sociodemographic subgroups (age groups, gender, educational attainment, and employment status), we examine potential differences in the levels of grit across these subgroups. Study 2 then sheds light on the criterion validity of this grit scale for to career success (income, job prestige, job satisfaction) and engagement (working overtime, participation in continuing professional development courses, attitudes toward lifelong learning). Using a representative subsample of gainfully employed respondents from the same large-scale survey, we test whether grit is incrementally associated with these outcomes over and above cognitive ability and sociodemographic characteristics, including educational attainment. We also illuminate the hotly debated [6,11] question as to whether this grit scale with its focus on perseverance possesses any incremental predictive validity over general Conscientiousness, of which grit is a facet [1,12]. Study 2 thus responds to recent calls for inquiries into the predictive power of grit for success in the occupational domain [6,11,13].”

This Introduction is not meant to go into much detail but only to provide an outline of the parts that follow. More in-depth explanations and justifications of the concepts and variables then follow in the revised literature review as well as in Study 1 and Study 2. 

[1.4] Moreover, in the background, other similar and opposite grit constructs should be considered, with the aim of highlighting points of convergence and divergence (for example, the concepts of perception of competence, vocation or procrastination).

We agree that other similar and opposite grit constructs could be considered. Some of them were already mentioned in the General Discussion. However, after careful consideration, we decided not to incorporate an in-depth discussion of additional constructs for three reasons. First, doing so would require considerable manuscript space. Given your concerns about manuscript length [1.2], we felt that we should not add to the length of the manuscript. Second, the present data did not include constructs as perception of competence or procrastination. Thus, we would not be able to test the relationships of grit to such constructs. Therefore, there would be little value in discussing these constructs at length. Third, the points of convergence and divergence of grit in relation to similar constructs were not the main focus of our study. Note, however, that the one construct to which grit is most closely related—namely Conscientiousness—is discussed in detail both in our literature review and empirically in Study 2.

[1.5] In line with the above, the objective is very generic and the analysis of results addresses issues that are not related to this objective.

The aims are very specific: Study 1 aims to shed light on the question as to how grit is distributed in the population and in relevant sociodemographic subgroups. For this purpose, it presents an in-depth evaluation of the psychometric properties of a short grit scale in large-scale representative adult sample. Study 2 aims to shed light on the question as to whether grit is associated with career success and engagement in an employed subsample; and whether these associations are incremental over cognitive ability, conscientiousness, and sociodemographic factors. 

In response to this comment and your previous ones, we have now stated these objectives of the two studies more clearly in the Introduction, and linked them more clearly to the literature review. We have also stated them more clearly in the Abstract in the Study 1 and Study 2 themselves. Additionally, in the literature review, each section clearly states how the issues raised in this section will be addressed by Study 1 or Study 2. 

[1.6] The results and discussion are considered together. The guidelines of the magazine propose that they constitute two distinct sections. Additionally, the length of these sections (7 pages for the first study and 9 for the second) makes reading and understanding the content very difficult.

Please allow us to clarify what appears to be a misunderstanding about the chosen structure of our paper. For two-study papers such as ours, it is common in psychology and the behavioral sciences to combine results and discussion of each individual study—and then have a general discussion at the end. This is precisely the structure we chose. We found it advantageous because the General Discussion gives an integrative summary the overall insights and contribution to the literature of the individual study.

Kindly note that the submission guidelines of PLOS ONE explicitly permit such a mixed Results/Discussion section, see https://journals.plos.org/plosone/s/submission-guidelines#loc-results-discussion-conclusions:

“These sections may all be separate, or may be combined to create a mixed Results/Discussion section (commonly labeled “Results and Discussion”) or a mixed Discussion/Conclusions section (commonly labeled “Discussion”). These sections may be further divided into subsections, each with a concise subheading, as appropriate. These sections have no word limit, but the language should be clear and concise.”

That said, we took very seriously your concern about the Results/Discussion sections being difficult to follow. Your comment stimulated us to shorten, rewrite, and restructure the Results/Discussion sections of both Study 1 and Study 2, resulting in considerably improved readability and clarity. Additionally, each subsection now ends with a paragraph that succinctly states the key insights of the respective analysis step (“In sum…”). We are very confident that these revisions resulted in substantially improved Results/Discussion sections, and we thank you for highlighting the issues with the previous versions.

[1.7] Some statements are poorly justified, for example, L. 147, "key publications"... It would be convenient to explain the reason why they are considered "key".

Thank you for pointing out. We have now clarified that we meant „most widely cited”.

[1.8] The introduction and conclusions should further emphasize the applications of the results as well as the consequences thereof.

Following this comment, we considerably revised the Conclusion section such that the implications of our results are clearer. We make clear that, according to our results, grit does provide added value because of its criterion validity for career outcomes. At the same time, we question whether future studies would be well advised to use grit scales—or whether they should better resort to more established and well-validated measures of Conscientiousness.

[1.9] The writing of the work should be improved. Here are some problems that need to be corrected:

On several occasions, acronyms are used without making explicit what they refer to (PIAAC; CASMIN ...).

In the revised manuscript, both terms are now defined upon their very first appearance in the paper: PIAAC refers to the Programme for the International Assessment of Adult Competencies. CASMIN refers to the Comparative Analysis of Social Mobility in Industrial Nations. It is a standard coding of educational attainment. 

[1.10] Line 283, a section with the title "Study 1" opens and the next line begins "Our aims in Study 2"

Corrected. Thank you.

[1.11] The first person of the plural is used, recommending the impersonal style. 

In line with several publication manuals in the social and behavioral sciences (e.g., the American Psychological Association’s (APA) manual, we prefer the first person plural because it is easier to read, shorter, and because the subject of each sentence is unambiguously clear.

[1.12] L. 63, citations 6 and 7 present considerable criticism, but this criticism is of a very different nature (one is a scientific article and the other comes from mass media). Likewise, these citations that appear on line 50 should also be assessed for their relevance in the context where they appear. In short, it is suggested to quote with more precision and criticism. 

This is true. In response to this comment, we clarified that one of the two sources is an article in the popular media: “[…] the grit construct has also drawn considerable criticism in research [6] and popular media [7]”. 

[1.13] The title of the work has been edited with words in capital letters. 

Changed to lower-case letters.

[1.14] The expression "Conscientiousness" is written throughout the text with the first capital letter. No other expression appears following this rule (?). 

Changed to lower-case letters throughout the manuscript.

[1.15] In summary, it is suggested to carefully rewrite the whole work, which may lead to a good article. Once the extension of the same is reduced, a review can be made more adjusted to the concreteness of what has been presented.

Thank you again for your comments. In responding to them, we feel that our manuscript is further strengthened.

 

Responses to Reviewer #2

The following technical concerns should be considered in the revision:

[2.1] Problem of short scales: Given large-scale omnibus studies as PIAAC, there is now a strong effort to use short scales in the hope of transferring properties from the parent scale to the short scale. Due to the fact that the manuscript is methodological oriented (“measurement invariance”), the general problems of short scales should be discussed as well (e.g., Smith et al., 2000). Especially, I wonder why a short scale of GRIT with 5 items was applied instead of the original scale in this validation study at the expense of a possible low reliability of the scale.

As suggested, we revised the Limitations section of the General Discussion. There, we discuss in greater detail the limitations of short scales. We cite the paper by Smith et al. (2000) that you kindly mentioned. We also explain why short scales were used in PIAAC-L (which we use as secondary data, namely because of time and questionnaire space constraints. Moreover, we call for studies replicating our findings with longer scales. The revised section reads as follows:

“The first limitation concerns the measures at our disposal in the PIAAC–L data. Because of time and questionnaire space constraints, both grit and conscientiousness were measured with short scales. This is a typical trade-off in multithematic large-scale surveys. Short scales are increasingly common and are often able to retain a considerable amount of the criterion validity of longer scales [63]. However, compared to longer scales, short scales are typically less reliable and sometimes content deficient, which may lead to attenuated and more variable criterion correlations [63,66,67]. Our use of latent-variable models ensured that the scales’ reliability was not an issue. However, the fact that the PIAAC-L data used only five items to measure grit—all four perseverance items from the Grit–S scale [16] but only one consistency item—implies that we were unable to recover the proposed two-facet structure of grit in our analyses [5]. Although the perseverance facet has emerged as more powerful predictor of success outcomes than the consistency facet or overall grit [11], future research using full-length grit scales that allow modeling and comparing both grit facets would be an important addition to our findings.”

As in the previous version, we then detail the more specific limitations concerning our analyses that resulted from using short scales for grit and conscientiousness.

[2.2] Structural equation modeling: The application of SEM was justified by low reliability (Omega=.63) of the scale (“Row 413 “…call for latent-variable models”). First, SEM is the method of choice in social sciences, if scales are prone to random measurement errors irrespective of the size of measurement errors. Second, reliability is a necessary, but not sufficient condition for validity despite any attenuation formulas. If the reliability is low, the validity is restricted in size. In my view this issues should be consider at least as part of the final discussion (limitations).

We fully agree with these statements. We have reworded several sentences in our manuscript to clarify some key points: First, low reliability of the scale score implies that the scale score should not be used for substantive analyses. Second, our use of latent-variable models ensured that criterion validity estimates were not attenuated by the low reliability of the scale score. 

[2.3] Response sets, social desirability: Rating scales are prone to response sets (e.g., acquiescence, middle-scale) and effects of social desirability, for example, the item “I am hard worker”. Such biases were mentioned in the manuscript (row 578), but they are, unfortunately, not considered in the statistical analysis. The revised manuscript should give some reasons, why the statistical analysis did not account for these possible distortions driven by the question format (Boeckenholt, 2017).

Thank you for raising this point. We are well aware of the issue of response styles or sets. We considered analytically accounting for response styles but decided that the data offered no statistically and substantively convincing way to do so. We now devote an entire new paragraph to this issue in the General Discussion, which reads as follows:

“Apart from cognitive ability, all measures used in this study were self-report measures. As such, they are prone to response styles such as acquiescence and socially desirable responding that can introduce common method bias and distort model fit, criterion correlations, and other covariance-based statistics (e.g., [68]). Future research should analytically account for such response styles, such as by using balanced scales to counter acquiescence, and by observer-rated or even objective measures of grit (e.g., [26]), conscientiousness, and career success and engagement.”

[2.4] Measurement model: Eventually, concepts of classical test theory were used to establish the measurement model. Rating scales were used, which are at best ordinal scaled. I wonder whether measurement models considering the ordinal character of the data are more adequate than models based on continuous scales, for example, an ordinal Rasch model with a threshold concept. In my view the reasons to favor concept of classical test theory toward the probabilistic test theory is worth to be shortly discussed in the revised manuscript.

It is correct that the scales are ordinal in nature. While we agree that item response theory (IRT) or item factor analysis (IFA) models would have been a possible alternative, we decided to follow classical test theory (CTT) and its extensions because they continue to be much more widely used and understood by applied researchers. We felt that this was more important than the potential advantages of IRT models, which tend to play out more in cognitive testing (e.g., computer-adaptive testing, equating different test forms, etc.).

Moreover, we explain in the Method section of Study 1 that—as per the simulation studies conducted by Rhemtulla and colleagues—the consequences of treating ordinal items as quasi-linear are typically minor with maximum likelihood estimation when five or more response categories are used. We used a robust maximum likelihood (MLR) estimator that also accounts for non-normality in the items’ distributions. 

Rhemtulla, M., Brosseau-Liard, P. É., & Savalei, V. (2012). When can categorical variables be treated as continuous? A comparison of robust continuous and categorical SEM estimation methods under suboptimal conditions. Psychological Methods, 17(3), 354–373.

[2.5] Statistical model comparison: The model comparison was mainly based on the goodness of fit indices, as CFI and residual indices as RMSEA. I wonder why the so-called information criteria as BIC (mentioned as aBIC? in the manuscript) were not used for model comparisons. With BIC the scalar measurement model can be clearly ruled out (Table 1). Given the low inter-correlations among the variables (Table S4), it could be expected that most of the models fit the data rather well despite low reliability. Model fit and reliability do not have to match (“First, it is possible that the scale is simply too short.“ Stanley et al., 2016, p. 984).

Apologies for overlooking this. Following this comment, we now explain in the Method section of Study 1 that aBIC refers to the sample-size adjusted BIC in Mplus (BIC or AIC would yield the same conclusions); and that lower aBIC values imply a better balance between fit and complexity (or parsimony). In the Results/Discussion section of Study 1, we then explicitly consider aBIC for the model comparisons. As you correctly point out, aBIC rules out the scalar invariance models, which is in agreement with the other fit indices, and favors the metric invariance models. 

We also fully agree with the second part of your comment. In the revised manuscript, we emphasize the point that reliability (internal consistency) is a function of the manifest scale score; and that, despite overall good model fit, we see the five-item grit scale in PIAAC-L in need for further improvement because of low reliability (omega) and low average variance extracted (AVE). Moreover, we caution against using the manifest scale score because of its low reliability and advocate the use of latent-variable modeling as a standard approach.

[2.6] Minor: I would like to point out that the title is only understandable for readers, who know what GRIT is. Perhaps the underlying construct is also interesting for other readers.

Very important hint, thank you. We have revised the title and added the paraphrase “effortful persistence” in parentheses. The same paraphrase also appears in the Abstract and Introduction when the term “Grit” is first mentioned. In this way, the topic of our study will be understandable for readers not yet familiar with the grit construct.

References

Smith, G. T., McCarthy, D., M., Anderson, K. G., (2000). On the sins of short-form development. Psychological Assessment, 12(1), 102-111.

Boeckenholt, U. (2017). Measuring response styles in likert items. Psychological Methods, 22(1), 69-83.

Stanley, L. M., Edwards, M. C. (2016). Reliability and Model Fit, 76(6), 976–985

Responses to Reviewer #3

Thank you for the opportunity to review the interesting manuscript. The authors conducted extensive and rigorous analyses on large-scale data of the German population to test the reliability and validity of the grit construct. I admire the thoughtful and thorough analytic approach taken by the authors, as they contribute compelling evidence regarding the value of grit in relation to other related constructs. Below I offer some suggestions to potentially improve the manuscript from its current state.

We are grateful for your positive evaluation of our manuscript. We have implemented your helpful suggestions, resulting in what we believe is a further improved paper. We detail our responses to your comments below.

[3.1] For the categorical variables (gender, education, employment), please indicate the frequencies of participants under each category.

We have added the missing frequencies in parentheses in the method section of Study 1. The values can also be calculated from the descriptive statistics in Table S1 and Table S3.

[3.2] In their evaluations of the factor loadings of grit items (for example, lines 391-395 and 741-749), the authors report that some items had “smaller loadings” while others were “substantial” based on the readings of coefficients. To add objectivity to the inference, authors may want to consider statistically testing the (in)equality of the coefficients. I believe there are several approaches to do this, like the one discussed in this paper:

Kwan, J.L.Y. & Chan, W. (2011). Comparing standardized coefficients in structural equation modeling: a model reparameterization approach. Behavior Research Methods, 43(3), 730-745. https://doi.org/10.3758/s13428-011-0088-6

In response to this comment, we clarified that some loadings were descriptively smaller than others, and that they were below widely accepted thresholds of what constitutes substantial loadings (typically λ ≥ .40). In line with your suggestion, we also tested the differences between each of the higher (>.50) and the two lower loadings for statistical significance using the approximately chi-square distributed Wald tests. Not surprisingly in view of the sample size, all of these comparisons were highly statistically significant. For example, comparing the third to the fourth item’s loading, χ2(1) = 46.866 p < 0.0001. We now briefly state this in the Results/Discussion of Studey 1 but do not report detailed results in order not to further extend manuscript length.

[3.3] In addition to the two models displayed in Figure 2, how would the model fit if you include grit and Conscientiousness in the model as independent first-order factors and assume an inter-factor correlation. I would expect that the model’s fit will be lower than that of model 2B, but the comparison would contribute to demonstrating whether, and to what extent, grit should be considered as a construct independent from Conscientiousness or as a facet of Conscientiousness.

You correctly surmised that such a model with two correlated first-order factors for grit and Conscientiousness would fit worse than the Bifactor-(S–1) model, χ2(19) = 155.282, p = .000, CFI = .926, TLI .89, SRMR = .039, aBIC = 44257.228. The correlation between grit and conscientiousness in the model was very high, r = 0.78, which will lead to multicollinearity in regressions using both constructs aas predictors. We now briefly report this in the Method section of Study 2:

[3.4] What specifically was the method of analyses for the results displayed in Figure 3? Were they simple regressions, or were they multiple regressions where you controlled for cognitive ability and conscientiousness when you regressed the DVs on grit?

Apologies if this was not sufficiently clear. These are multiple regressions in which cognitive ability, conscientiousness (in Model 2B) and all covariates are controlled. We now highlight in the caption of Fig. 3 that, “All associations are controlled for the covariates shown in Tables 3 and 4.”

[3.5] In line 864, the authors report that the “grit factor was unrelated to job prestige,” while the result displayed in Table 4 indicate the particular coefficient was significant at least at the .05 level. Perhaps the authors were referring to the results of Figure 3, but it is difficult to evaluate the effects just from the visualization of the CIs. Please indicate specific CI values for the effects depicted in Figure 3.

You are absolutely right. Thank you for paying such close attention. We now explain that there was an association with job prestige—but that the 95% CI bordered zero, and the coefficient was only significant at p < 0.05. We calculated effect sizes for the association. 

We also followed your very sensible suggestion to report the exact 95% CI values. Because we could not integrate them in the Figures without making it very hard to read, we report them in Tables 3 and Table 4 (instead of standard deviations). This adds considerably to transparency. 

[3.6] According to Study 2, grit, when treated as a residual factor after controlling for Conscientiousness, retains its associations with variables such as income, job satisfaction, and hours spent working overtime, but ceases to predict two career engagement variables (i.e., number of CPD courses taken, learning orientation). I find this an interesting set of findings and wish for the authors to elaborate on what the results imply. For instance, the findings seem to contradict the existing view that grit differs from general Conscientiousness in that it “emphasizes long-term goal striving” (line 246 of manuscript). The results of the present study suggest that general Conscientiousness more strongly predicts employees’ orientation toward long-term career development, while grit holds predictive power over people’s engagement with the job at hand (i.e., how long they persist with the task, how affectively committed they are with the current job). I would like to see a discussion on what the findings add to the debate about how grit should be conceptualized.

We find this a very interesting observation. We picked it up in the Results/Discussion section of Study 2, where we now state:

“Overall, conscientiousness was less consistently related to career success than the residual grit facet—but more consistently related to career engagement. As an anonymous reviewer noted, the fact that grit hardly predicted engagement after removing the variance it shares with conscientiousness may call into question whether what distinguishes the constructs is really grit’s focus on long-term goal striving (i.e., engagement).”

At the same time, we were careful not to overstate this interpretation. In our view, the models do not rule out that grit relates to long-term striving because (a) grit was still related to income and job prestige, which are commonly seen as the result of long-term efforts in one’s career; and (b) Model B does not pertain to grit “per se” (as Model A does) but more specifically to the part of grit that is independent of conscientiousness. Model B’s test of grit’s criterion validity is, thus, much more conservative. 

[3.7] Format for reference citation was inappropriate throughout the results section of Study 1, and in several other places as well (lines: 64, 375). 

Corrected. Thank you for pointing out.

[3.8] Unidentified footnotes appear in lines 48 and 149. Also please cite the TED talk mentioned in line 149. 

Because PLOS ONE does not allow footnotes, we integrated all previous footnotes in the text. We removed unidentified footnotes. 

As requested, we also added a citation and URL to the TED talk.

[3.9] Please check the sentence structure for lines 692-695. 

The wording was indeed confusing. We now state more simply that, 

“Model B is a Bifactor-(S–1) model [46] in which the three conscientiousness items and the five grit items all load on a conscientiousness factor; whereas the five grit items, but not the three conscientiousness items, load on a grit facet factor.”

[3.10] Although the manuscript in overall was well-written, I found typographical errors throughout. Please thoroughly review the manuscript for errors. Below, I list ones I noticed with their line numbers: 

95: such *as* lower socioeconomic strata / 183: grit was be associated / 285: Study 2 / 524: was .83. graphs but… / 560: how relates / 566: (3) / 599: these any / 626: though 827: pattern is suggests / 967: old wine in new *models / 992: wind out / 1004: helped use

Thank you for your careful reading and for drawing our attention to these typos. We corrected them. In addition, we subjected our entire manuscript to another round of proofreading. We hope that all typos have now been eliminated.

---

## [Decision Letter · Decision Letter 1]

23 Oct 2019

Grit (effortful persistence) can be measured with a short scale, shows little variation across socio-demographic subgroups, and is associated with career success and career engagement

PONE-D-19-18723R1

Dear Dr. Lechner,

We are pleased to inform you that your manuscript has been judged scientifically suitable for publication and will be formally accepted for publication once it complies with all outstanding technical requirements.

With kind regards,

Angel Blanch, Ph.D.

Academic Editor

PLOS ONE

Additional Editor Comments (optional):

Reviewers' comments:

Reviewer's Responses to Questions

**Comments to the Author**

1. If the authors have adequately addressed your comments raised in a previous round of review and you feel that this manuscript is now acceptable for publication, you may indicate that here to bypass the “Comments to the Author” section, enter your conflict of interest statement in the “Confidential to Editor” section, and submit your "Accept" recommendation.

Reviewer #1: All comments have been addressed

Reviewer #2: All comments have been addressed

Reviewer #3: All comments have been addressed

2. Is the manuscript technically sound, and do the data support the conclusions?

Reviewer #1: Yes

Reviewer #2: Yes

Reviewer #3: (No Response)

3. Has the statistical analysis been performed appropriately and rigorously? 

Reviewer #1: Yes

Reviewer #2: Yes

Reviewer #3: (No Response)

4. Have the authors made all data underlying the findings in their manuscript fully available?

Reviewer #1: Yes

Reviewer #2: Yes

Reviewer #3: (No Response)

5. Is the manuscript presented in an intelligible fashion and written in standard English?

Reviewer #1: Yes

Reviewer #2: Yes

Reviewer #3: (No Response)

6. Review Comments to the Author

Reviewer #1: (No Response)

Reviewer #2: Thank you very much for your great revision job on the manuscript. I am very impressed by the corrections that you made and I am looking forward to the publication.

Reviewer #3: (No Response)

7. PLOS authors have the option to publish the peer review history of their article (what does this mean?). If published, this will include your full peer review and any attached files.

Reviewer #1: No

Reviewer #2: No

Reviewer #3: No

---

## [Editor Report · Acceptance letter]

19 Nov 2019

PONE-D-19-18723R1 

Grit (effortful persistence) can be measured with a short scale, shows little variation across socio-demographic subgroups, and is associated with career success and career engagement 

Dear Dr. Lechner:

I am pleased to inform you that your manuscript has been deemed suitable for publication in PLOS ONE. Congratulations! Your manuscript is now with our production department. 

With kind regards,

on behalf of

Dr. Angel Blanch 

Academic Editor

PLOS ONE